

# The changing mass of the Antarctic Ice Sheet during ENSO-dominated periods in the GRACE era (2002-2022)

John Bright Ayabilah [1,2], Matt King[1,2], Danielle Udy[2], Tessa Vance[3]

[1]School of Geography, Planning, and Spatial Science, University of Tasmania, Hobart 7001, Tasmania, Australia
[2]The Australian Centre for Excellence in Antarctic Science, Institute for Marine & Antarctic Studies, University of Tasmania, Hobart 7001, Tasmania, Australia
[3]Australian Antarctic Program Partnership, Institute for Marine & Antarctic Studies, University of Tasmania, Hobart, TAS, 7001, Australia

*Correspondence to*: John Bright Ayabilah (johnbright.ayabilah@utas.edu.au)

**Abstract.** Large-scale modes of climate variability significantly influence Antarctic Ice Sheet (AIS) mass change. Improved understanding of the relationship between these climate modes and AIS mass change can help reduce uncertainties in future ice mass estimates and its contribution to sea level rise. However, the spatiotemporal patterns of AIS mass variation driven by El Niño Southern Oscillation (ENSO)-induced atmospheric circulation remain unclear. Here, we investigate AIS variability during different ENSO periods using Gravity Recovery and Climate Experiment (GRACE) observed mass changes over the period 2002 to 2022. The results show strong event-to-event spatial variability in how the ENSO teleconnection manifests over the AIS. These differing spatial patterns are primarily driven by changes in the Amundsen Sea Low (ASL) strength, location, and extent, which alter circulation patterns and moisture flow in West Antarctica. In East Antarctica, ice mass variability is largely influenced by the positioning of cyclonic and anticyclonic anomalies, primarily driven by the Southern Annular Mode (SAM); however, ENSO signals are also present. In both East and West Antarctica, this study shows that the spatial impact of any given ENSO event, as derived using standard tropical atmospheric metrics (Sea Surface Temperature (SST) and pressure anomalies), and its influence on the ASL and Southern Ocean circulation can be equally (and in some cases more) important to AIS variability. GRACE provides an opportunity to understand event-scale ENSO precipitation independently of numerical models.

## 1. Introduction

The drivers of inter-annual to decadal Antarctic Ice Sheet (AIS) mass variability are complex and not yet fully understood. External factors, such as episodic extreme precipitation events often linked to atmospheric rivers (Wille et al., 2021), and internal factors, including ice dynamics (IMBIE Team, 2018), both contribute to these variations. Understanding the mechanisms underlying AIS mass change and variability is critical for improving future projections of ice mass changes and the Antarctic contribution to sea level rise.

The main determinants of the net AIS mass balance (MB) are ice discharge (D) from the continental margins of Antarctica and Surface Mass Balance (SMB). SMB is further defined as accumulating precipitation onto the ice sheet, minus runoff, sublimation/evaporation and blowing snow erosion. The fluctuation of the AIS mass balance and its subsequent contribution to sea level rise are based on the difference between ice discharge and SMB (i.e., MB = SMB – D). The AIS SMB exhibits high variability on inter-annual to decadal timescales, (Kim et al., 2020;





Medley and Thomas, 2019; Van De Berg et al., 2006). Precipitation variability, driven by atmospheric circulation,
is a key determinant of Antarctic SMB and, over a wide range of timescales, including interannual to decadal, is
closely linked to modes of climate variability (Kim et al., 2020).
The Southern Annular Mode (SAM) is the dominant mode of extratropical variability in the Southern Hemisphere.
SAM signal in Antarctic precipitation is regionally dependent and affects different regions of Antarctica in distinct
ways (Marshall et al., 2017). During the positive phase of SAM, the mid-latitude westerly wind belt contracts
poleward, with a reduction in net precipitation across Antarctica (Marshall et al., 2017; Medley and Thomas,
2019). Conversely, the negative phase of SAM, is associated with increased net precipitation over the continent
(Medley and Thomas, 2019; Marshall et al., 2017). Regionally, the contraction of the storm track during positive
SAM strengthens the westerlies around 60° S, enhancing moisture transport to the coastal regions of West
Antarctica and the western Antarctic Peninsula, which increases precipitation. In contrast, the contraction of the
westerlies reduce moisture transport to the interior of East Antarctica, decreasing precipitation, with the reverse
pattern occurring during negative SAM (Medley and Thomas, 2019; Marshall et al., 2017). However, SAM related
circulation patterns are not stationary and vary over decades, meaning the regional impacts may shift over time
(Marshall et al., 2013).
The El Niño Southern Oscillation (ENSO) is the dominant mode of inter-annual climate variability globally (2–
7-year timescales) and is defined by variations in sea surface temperature (SST) anomalies in the tropical Pacific
(Mcphaden et al., 2006). The ENSO pathway to Antarctica is modulated by the Amundsen Sea Low (ASL)-
poleward end of the Rossby wave train (Hoskins and Karoly, 1981). These interactions create high-pressure
anomaly over the Amundsen-Bellingshausen sector (ABS) during El Niño and low-pressure anomaly during La
Niña conditions (Turner, 2004; Hoskins and Karoly, 1981). The ASL represents a climatological area of low
pressure in the South Pacific and is a key component of the nonzonal climatological circulation (Raphael et al.,
2016b). The teleconnection between ENSO and the ASL is strongest during the austral spring (September-
November; SON) but exerts influence throughout the year (Schneider et al., 2012; Clem and Fogt, 2013; Fogt et
al., 2011). The strength, extent, and location of the ASL shows significant variability during different ENSO
phases and individual ENSO events, resulting in varying atmospheric circulation patterns that strongly influences
moisture and temperature distribution in West Antarctica (Raphael et al., 2016b; Hosking et al., 2013). The impact
of ENSO on East Antarctica through the ASL is not fully clear (Zhang et al., 2021; King et al., 2023).
The impact of ENSO on Antarctic climate is modulated by the phase of SAM, with the signal amplified when
SAM and ENSO are atmospherically in phase (positive SAM/La Niña or negative SAM/El Niño) and reduced
when they are atmospherically out of phase (positive SAM/El Niño or negative SAM/La Niña) (Clem et al., 2016;
Fogt et al., 2011). Positive SAM and La Niña conditions are associated with a deepening (i.e. lower pressure
anomaly) ASL, while negative SAM and El Niño conditions weaken the ASL, and influence its longitudinal shift
(Raphael et al., 2016b; Hosking et al., 2013). The deepening of the ASL induces continental wind outflow on its
western flank, reducing precipitation and SMB in West Antarctica, whereas a weakened ASL leads to onshore
winds that enhance precipitation and SMB (Zhang et al., 2021; Li et al., 2022a). The longitudinal shift of the ASL
modifies these impact zones.

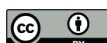



The spatial patterns and magnitude of AIS mass variability due to large-scale modes of climate variability remain
unclear. Studies on the role of ENSO in Antarctic climate have mostly focused on precipitation derived from
reanalysis products or modelled SMB data (e.g., Medley and Thomas, 2019; Clem et al., 2016; Clem and Fogt,
2013; Fogt et al., 2011). Only a few studies have examined the relationship between large-scale modes of climate
variability and recent observed ice mass variation using Gravity Recovery and Climate Experiment (GRACE)
observed AIS ice mass change time series on timescales ranging from months to decades (e.g.,Bodart and
Bingham, 2019; Zhang et al., 2021; King et al., 2023). Most of these studies have focused on single strong ENSO
events, such as the 2015-2016 El Niño (Bodart and Bingham, 2019), or on the mean impact of ENSO on the AIS.
The GRACE mission, launched in 2002, has contributed to our understanding of the redistribution of mass within
the Earth system, which is useful for observing changes of the Greenland and Antarctic ice sheets (Tapley et al.,
2004; Shepherd et al., 2012). GRACE-observed ice mass variability is related to atmospheric circulation-driven
snow accumulation and variation in ice discharge (Diener et al., 2021), with atmospheric variability dominating
over interannual timescales (King et al. 2023). Studies of ENSO's impact on AIS using GRACE-observed ice
mass changes show that different ENSO events result in varying climatic and surface weather effects, leading to
different spatial patterns of AIS mass variability. Bodart and Bingham (2019) demonstrated that during the 2015-
2016 El Niño, the Antarctic Peninsula and West Antarctica gained mass, while East Antarctica experienced a
reduction in mass. This spatial pattern is also consistent over a longer period, in line with Zhang et al. (2021) who
found similar correlations. They observed a bipolar spatial pattern: during El Niño events, there was a mass gain
over the Antarctic Peninsula and West Antarctica and a mass loss over East Antarctica, while the pattern reversed
during La Niña events. The bipolar spatial patterns are consistent with the results of King et al. (2023), based on
a GRACE analysis for the period 2002-2021, and King and Christoffersen (2024), which used GRACE and
altimetry data (2002-2020), despite differences in approaches and study periods. However, other studies have
suggested that specific ENSO events and types of ENSO events have distinct impacts on Antarctic SMB that are
not limited to a bipolar pattern (e.g., Macha et al., 2024; Sasgen et al., 2010).
This study aims to investigate the spatial patterns of ice mass change and the driving atmospheric circulation
conditions during various ENSO-dominated periods, as observed in GRACE-derived AIS mass variations
between 2002 and 2022. The results indicate that no two ENSO events have the same net effect on Antarctic ice
mass, especially at regional scales, and the bipolar spatial pattern observed in earlier studies is not consistent
across all ENSO events. This variability suggests that the ENSO signal in the AIS is shifted from its background
pattern depending on event-specific atmospheric and oceanic factors.
**2. Data and Methods**
**2.1. AIS mass change**
We used the GRACE and GRACE Follow On data provided by the GFZ German Research Centre for Geosciences
(Landerer et al., 2020). The GRACE Follow-On mission, launched in May 2018, succeeded the GRACE mission,
which was decommissioned in October 2017 due to battery and fuel problems. This gap between the GRACE and
GRACE Follow-On missions resulted in the loss of data from July 2017 and May 2018. Our analysis involved
GRACE data spanning from April 2002 to Dec 2022 without gap filling. We used COST-G RL-01 V0003 50km





gridded products with approximately monthly temporal sampling, but note that GRACE data have an underlying
spatial resolution of ~300km (Sasgen et al., 2020; Dahle et al., 2024).
The various available GRACE data products differ based on the processing methods and background models used.
The gridded mass change product adopted here is initially derived by solving for spherical harmonic coefficients
and then computing mass anomalies for each grid cell across the entire ice sheet using tailored sensitivity kernels
that minimise both GRACE and leakage error (Groh and Horwath, 2016). Within this product, glacial isostatic
adjustment is corrected using the ICE6G_D model (Richard Peltier et al., 2018), although this has no bearing on
non-linear variability as studied here. Atmospheric and oceanic effects on mass redistribution are also modelled
as are spherical harmonic degree-1 terms based on the approach of Swenson et al. (2008). Further details about
the GRACE time series, post-processing techniques, and quality assessment can be found in Dahle et al. (2019).
It is worth noting that the GRACE-observed ice mass change time series is affected by systematic errors associated
with the GRACE orbital geometry and small unmodelled errors, evident in the (largely north-south) striping
pattern observed in some of the ice mass change results.
We focus our analysis on the ENSO signal in ice mass variation during different ENSO-dominated periods. First,
we removed short-term signal fluctuations in the GRACE data by applying a 7-month moving median smoother
to the GRACE time series. Since our focus is on GRACE-observed ice mass variability, we subtracted the linear
trend at each grid point, estimated using ordinary least squares over the data span. This effectively produces mass
anomalies with respect to the climatology of the entire GRACE period.
To understand the relationship between ice mass changes and ENSO-dominated periods, we computed the rate of
ice mass change for each identified ENSO-dominated period. These rates represent the impact of ENSO during
each ENSO-dominated period. We calculated the rates for each grid cell of the gridded GRACE ice mass anomaly
data and generated spatial patterns of ice mass trends for each ENSO-dominated period.
**2.2. Climate indices**
To characterise ENSO variability, we used the Niño3.4 index, one of several metrics that measures the strength
and phase of ENSO based on sea surface temperature anomalies in the central and eastern tropical Pacific. This
index is obtained by tracking the running five-month mean SST based on the HadISST record over 5°N–5°S,
170°W–120°W (Rayner et al., 2003) and is shown in Fig. 1a. It is provided by the Climate Prediction Centre
(CPC) of the National Oceanic and Atmospheric Administration (NOAA) and can be accessed at
https://psl.noaa.gov/data/timeseries/month/Nino34/. The Niño3.4 temperature anomalies are standard for
detecting and monitoring ENSO events but cannot differentiate between eastern and central ENSO events. We
used the Niño3.4 index because our focus was on the spatial variability in AIS mass during all ENSO events,
rather than differentiating between eastern and central ENSO events.
For SAM, we used the station-derived index from Marshall (2003), available at http://www.nerc-
bas.ac.uk/icd/gjma/sam.html, and shown in Fig. 1a. This index is based on the zonal pressure differences at 12
stations located between 40 ° S and 65 ° S.



To investigate the potential linkage between large-scale climate variability and ice mass variation, we
cumulatively summed all the climate indices as shown in Fig 1b, c. The AIS mass reflects the compound effect
of surface mass fluxes over time. The cumulative mass flux observed by GRACE reflects the cumulative climate
indices (King et al., 2023) as opposed to raw indices, which relate to mass flux. These cumulative indices are also
captured by modelled cumulative SMB (Kim et al., 2020; Diener et al., 2021). The alternative approach is to
difference GRACE data in time, but this inflates the GRACE noise and reduces the lower frequency signal and is
hence undesirable (King et al., 2023).
To identify ENSO signatures in the GRACE data, we first identified El Niño- and La Niña-dominated periods
based on the cumulative summed indices, which essentially act as a low-pass filter of the raw indices. The
cumulative summed indices were derived from anomalies relative to their climatological mean using a reference
window of 1971-2000. This period is a well observed period before the commencement of GRACE and is the
same as that chosen by King et al. (2023). After the indices were normalised using the mean and standard deviation
computed within the reference window, the normalised indices were restricted to the GRACE period, cumulatively
summed, detrended, and renormalised.
In this study, we defined El Niño-dominated periods as those where the positive phase of ENSO dominates the
negative ENSO phase until a positive peak in the cumulative index is reached. Conversely, La Niña-dominated
periods are defined as those in which the negative phase dominates until a negative peak is reached. In a
cumulatively summed index, these are expressed as sustained periods of positive (El Niño) or negative (La Niña)
slope. Based on this criterion, we identified four El Niño-dominated periods over the GRACE time steps: 2002-
2005, 2009-2010, 2014-2016, and 2018-2020 (Fig. 1d). An equal number of La Niña-dominated periods were
found, covering 2007-2009, 2010-2014, 2016-2018, and 2020-2022. The strength of the expression of the ENSO
signal in the Antarctic climate is modulated by the phase of SAM (Fogt et al., 2011). During the 2002-2005 El
Niño-dominated period, the cumulative SAM index was dominated by negative SAM until around 2008
(atmospherically in phase El Niño/-SAM). After 2008, the cumulative SAM index exhibited no notable trend,
indicating a neutral phase. During the 2014-2016 El Niño, cumulative SAM and ENSO indices were
atmospherically out of phase (El Niño/+SAM). SAM shifted to a neutral state during the 2016-2018 La Niña.
SAM and ENSO were atmospherically in phase during the 2018-2020 El Niño (El Niño/-SAM) and 2020-2022
La Niña (La Niña/+SAM), which is notable as the only time positive SAM and La Niña co-occurred over the
GRACE period (Fig. 1d, e).





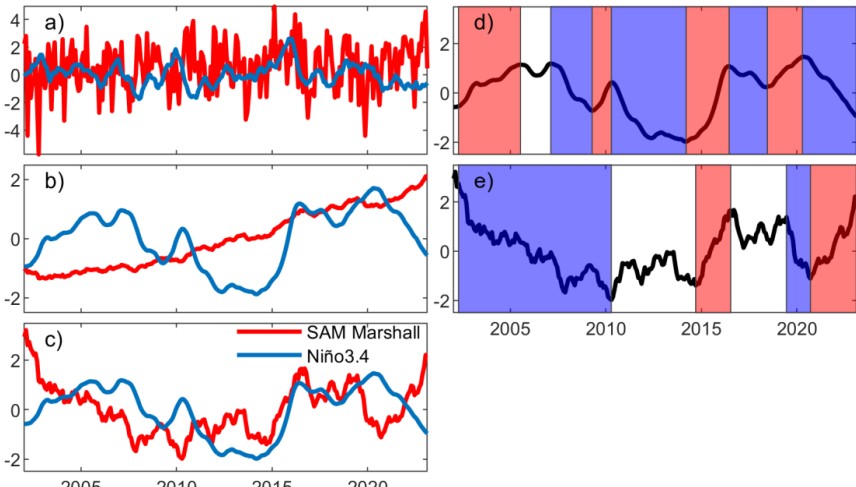

**Figure 1.  Monthly climate indices of SAM (Marshall et al) and Niño3.4 from 2002-2022. Normalised raw indices are shown in a), b) shows the cumulatively summed normalised raw indices after which it is renormalised. The signals of the cumulatively summed indices after removing the linear trend are shown in c). Positive peaks in cumulatively summed Niño3.4 follows El Niño dominated state and negative peaks follows La Niña dominated state. d) cumulatively summed ENSO index, red and blue shaded areas represent El Niño- and La Niña-dominated periods, e) cumulatively summed SAM index (Marshall, 2003), red and blue shaded areas represent SAM positive and SAM negative dominated periods. Neutral dominated periods are represented by white shading.**

## 2.3. SMB model outputs

We used modelled SMB output from the Regional Atmospheric Climate Model RACMO2.3p2 model (Van Wessem, 2023). This model has a horizontal resolution of 27 km and a vertical resolution of 40 atmospheric levels. This version of SMB model output is forced by ERA5 reanalysis data at its lateral and ocean boundaries, with data available from 1979 onward. For our study, monthly SMB values truncated to the GRACE period were used, covering Apr 2002 to Dec 2022. To compare with GRACE data, we computed anomalies relative to the 2002-2022 mean and then cumulatively summed them to obtain cumulative SMB anomalies in units of kg m$^{-2}$. These anomalies were then interpolated to match the GRACE grid spacing and time steps. We detrended the cumulative SMB and performed a regression analysis on these anomalies for each defined ENSO-dominated period.

## 2.4. Reanalysis climate data

To explore the potential climatic forcing during an ENSO-dominated period, we examined monthly mean ERA5 reanalysis model 10 m winds and sea level pressure from 2002 to 2022, with a resolution of 0.25 ° by 0.25 ° (Hersbach et al., 2020). Anomalies of 10 m zonal and meridional wind components, as well as sea level pressure, were computed for each grid cell relative to the mean over the GRACE period, for all regions south of 40° S. We then computed anomaly composite means for each ENSO-dominated period.



**2.5. Definitions of events, periods and anomaly interpretations used in this study**

We acknowledge that we use multiple terminologies in this study to define both our results, and when comparing to the literature. For example, we use the term 'El Niño- or La Niña-dominated period' to define the periods of time we define using our cumulatively summed index. In contrast, when comparing to or describing other literature, we use the term 'El Niño/ La Niña event' which refers to the peak phase of ENSO events. We also describe anomalies from the mean over the GRACE period. For the purposes of this study, the pressure and wind fields, as well as SMB and GRACE mass change, depicted in the figures represent anomalies from the climatology for each relevant variable. That is, for a given wind and pressure map, the fields depict wind and pressure anomalies against the 2002-2022 mean (the GRACE data period). Specifically, positive anomalies over the Antarctic continent reflect a strengthening of the mean Antarctic high pressure system while negative anomalies reflect a weakening of the high pressure (not the presence of a low-pressure system). Similarly, positive SMB and GRACE anomalies represent an increase in mass, whereas negative indicate a reduction in mass relative to the climatology.

**3. Results**

**3.1 Ice mass change**

We start by examining the long-term trend in AIS mass change over the GRACE observational period (Fig. 2). The spatial pattern reveals strong regional variability, with areas of both positive and negative mass anomalies. In West Antarctica, ice mass loss is most pronounced in the Amundsen Embayment and Bellingshausen Sea sectors, where accelerated ice discharge is well documented (Rignot et al., 2019; Gardner et al., 2018). The East Antarctic ice sheet shows mass gain across Dronning Maud Land (and through to Enderby Land), whereas the Wilkes Land section has experienced a decline in mass.

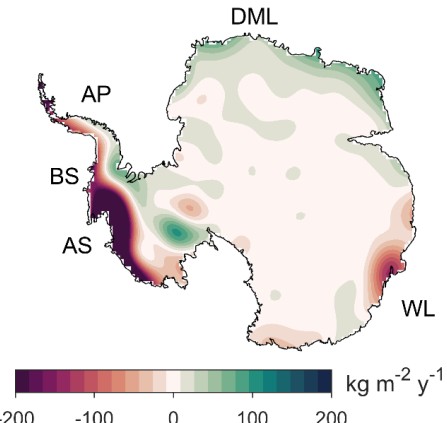

**Figure 2. AIS linear trend in ice mass change (2002-2022) based on GRACE data from a univariate regression. key Antarctic regions of interest are highlighted: Antarctic Peninsula (AP), Bellingshausen Sea (BS), Amundsen Sea (AS), Wilkes Land (WL), and Dronning Maud Land (DML).**





Figure 3 presents the regression results of cumulatively summed anomalies in climate variables (sea level pressure
and 10 m winds) and SMB, along with GRACE-derived ice mass change anomalies, against the cumulatively
summed Niño3.4 index. All variables were detrended before regression to focus on the variability. The results
show that ENSO influences atmospheric circulation over Antarctica (Fig. 3a), driving short-term fluctuation in
AIS mass around the overall trend. ENSO-induced changes in meridional flow regulate precipitation patterns,
making SMB a primary driver of AIS mass variability. Since SMB directly influences ice mass changes, this
results in spatially coherent patterns between SMB and GRACE-derived ice mass change (Fig. 3b–c). A
substantial portion of the ENSO and SAM signals in GRACE-observed ice changes can be linked to SMB
variability (Kim et al., 2020; King et al., 2023).

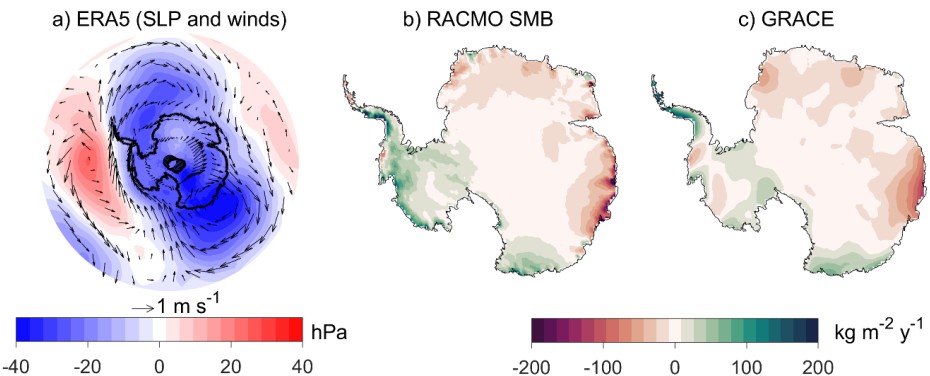

**Figure 3. Map of terms for cumulatively summed sea level pressure and 10 m wind anomalies from ERA5
reanalysis, cumulatively summed RACMO SMB anomalies, and GRACE ice mass change anomalies
when regressed against cumulatively summed Niño3.4. All variables are detrended prior to regression.**

The positive SMB anomalies in West Antarctica and negative anomalies in East Antarctica align with findings
indicating increased precipitation during El Niño and reduced precipitation during La Niña (Zhang et al., 2021;
Zhan et al., 2021). During El Niño events, strengthened onshore winds over West Antarctica enhance SMB, while
intensified offshore winds over East Antarctica reduce SMB. Conversely, during La Niña events, the circulation
pattern reverses, with increased moisture transport into East Antarctica and reduced onshore winds over West
Antarctica. As a result, SMB increases in East Antarctica while decreasing in West Antarctica. In West Antarctica,
the SMB signal differs from GRACE-derived ice mass changes, whereas in East Antarctica, the two signals are
more closely aligned. This suggests that SMB variability is the primary driver of ice mass changes in East
Antarctica but not necessarily in West Antarctica. The discrepancy may stem from the near-instantaneous
response of ice dynamics to ENSO-driven oceanic forcing and/or mismodelled SMB (IMBIE Team, 2018; Rignot
et al., 2019), with the latter being more likely (King and Christoffersen, 2024).
Given that no two ENSO events are identical, and the results in Fig. 3 reflect the mean AIS response—potentially
biased toward stronger ENSO events—we next examine AIS mass change, SMB variability, and the atmospheric
circulation driving these changes during different ENSO-dominated periods (Figs. 4 and 5). The results reveal



distinct spatial patterns of ice mass change associated with individual El Niño and La Niña events. We remind the
reader that the GRACE signal is more reliable in the coastal regions and less reliable in the interior, where inherent
systematic errors in GRACE measurements in the form of north-south striping are more pronounced.

**3.2. El Niño-dominated periods**

We analyse the variations in atmospheric circulation, SMB, and the resulting ice mass change during each defined
El Niño-dominated period throughout the GRACE observational record (Fig. 4). Across the Antarctic continent,
pressure anomalies indicate either intensification (high-pressure) or weakening (low-pressure) of the Antarctic
high (Fig. 4a–b). These variations align with the cumulatively summed SAM indices (Fig. 1e), where high-
pressure anomalies correspond to prolonged negative SAM phases, and low-pressure anomalies coincide with
prolonged positive SAM phases.








**Figure 4. Composite anomaly maps of sea level pressure and 10 m wind from ERA5 reanalysis, representing climatic conditions during El Niño-dominated periods relative to the climatology of the GRACE period (2002–2022). Sea level pressure anomalies are shown as shaded regions (hPa), while wind anomalies are indicated by reference vectors (m s⁻¹). The rate of change in Antarctic SMB is derived from the RACMO2.3p2 model (kg m⁻² y⁻¹), and the rate of surface mass change is obtained from GRACE data (kg m⁻² y⁻¹). SMB and GRACE maps illustrate trends in ice mass variability for each identified El Niño-dominated period. The GRACE signal is more reliable in the coastal regions and less reliable in the interior, where GRACE systematic error in the form of north-south striping is more evident.**

### 3.2.1. Atmospheric circulation and mass anomalies in West Antarctica

El Niño-dominated periods are characterised by a high-pressure anomaly in the Pacific sector, representing a weakened and/or shifted ASL rather than an actual high-pressure system (Fig. 4a–b). The position and strength of this high-pressure anomaly varies significantly within each El Niño-dominated period, influencing meridional circulation, thus driving distinct spatial patterns in SMB (Fig. 4e–h) and mass change (Fig. 4i–l).

Considering West Antarctica as two regions, the Amundsen Sea sector and the Antarctic Peninsula, SMB and mass change anomalies during the 2002–2005, 2009-2010, and 2014-2016 El Niño-dominated periods were of different signs but broadly uniform across both sectors (Fig. 4e–g, i–k). Conversely, the 2018–2020 period lacked this uniformity, displaying strong negative anomalies in the Peninsula and strong positive anomalies in the Amundsen Sea sector (Fig. 4h, l). During the earlier three El Niño-dominated periods, the high-pressure anomaly over the Pacific sector extended from the Amundsen to the Bellingshausen Sea (Fig. 4a–c). For the 2018–2020 El Niño-dominated period, the high-pressure anomaly in the Pacific was weaker (closer to climatology) and mainly located in the Bellingshausen Sea (Fig. 4d). These spatial variations demonstrate how the high-pressure anomaly's position significantly influences regional SMB and mass change patterns by controlling meridional circulation.

The Amundsen Sea sector exhibits consistent positive SMB (Fig. 4e, g–h) and ice mass anomalies (Fig. 4i, k–l) during three out of four El Niño-dominated periods (2002–2005, 2014–2016, and 2018–2020), despite variations in the location and strength of the high-pressure anomaly in the Pacific (Fig. 4a, c–d). Positive SMB and ice mass anomalies predominantly affect the Amundsen Embayment during these periods, with the most pronounced anomalies observed in GRACE data during 2002-2005 (Fig. 4i) and in both SMB and GRACE data during 2018-2020 (Fig. 4h, l).

The 2014–2016 El Niño-dominated period, which encompasses the extreme 2015-2016 El Niño event (Bodart and Bingham, 2019) within the GRACE observation period, coincided with weaker positive anomalies in the Amundsen Sea sector compared to the other periods (Fig. 4g, k). This period uniquely occurred out of phase with SAM (Fig. 1e), as evidenced by low-pressure anomalies over the continent that weakened the Antarctic high. During this period, the high-pressure anomaly in the Pacific shifted northward (Fig. 4c), with northerly wind anomalies flow over the Ross and Amundsen Seas corresponding to observed positive anomalies. A low-pressure anomaly position between the Ross and Amundsen Seas, contributed to onshore winds and positive anomalies (Fig. 4c).





The 2009–2010 El Niño-dominated period represents a notable exception to the other three periods in regard to
SMB and ice mass change anomalies in the Amundsen Embayment. Unlike other periods, negative anomalies
appeared in the Amundsen Embayment in both SMB and GRACE data (Fig. 4f, j). The characteristic northerly
wind flow typically associated with the other El Niño-dominated periods in the Amundsen sector was absent.
Instead, a high-pressure anomaly positioned further west than in the other three periods (between the Amundsen
and Ross Seas) generated anomalous southerly winds, resulting in offshore flow from the continent's interior (Fig.
4b).
The Antarctic Peninsula exhibits two distinct mass variability responses during El Niño-dominated periods (Fig.
4). The 2002–2005 and 2014–2016 El Niño-dominated periods show similar responses, with the Peninsula
experiencing positive SMB (Fig. 4e, c) and GRACE (Fig. 4i, k) anomalies supported by a high-pressure anomaly
in the Pacific driving northerly winds across the region (Fig. 4a, c). Note that the 2002-2005 SMB anomaly is
only marginally positive (Fig. 4a). In contrast, during the 2009–2010 and 2018–2020 El Niño-dominated periods,
southerly wind anomalies prevailed (Fig. 4b, d), resulting in a negative SMB (Fig. 4f) and ice mass anomaly (Fig.
4j) over much of the Peninsula. Also, during the 2009–2010 period, a strong low-pressure anomaly over the
Weddell Sea induced northerly winds along the eastern Peninsula (Fig. 4b), creating localized positive SMB and
ice mass anomalies (Fig. 4f, j).

### 3.2.2. Atmospheric circulation and mass anomalies in East Antarctica

El Niño events have been linked to negative mass anomalies in the East Antarctic Ice Sheet (King et al., 2023; Li
et al., 2022a), consistent with our earlier findings (Fig. 3b–c). The 2014–2016 and 2018–2020 El Niño-dominated
periods align with this general pattern, showing mostly negative anomalies in SMB (Fig. 4g–h) and GRACE data
(Fig. 4k–l) across East Antarctica. However, our analysis reveals that the relationship between El Niño and the
East Antarctic Ice Sheet is not limited to negative mass anomalies, with varying responses observed across the
Atlantic and Indian Ocean sectors.
In Dronning Maud Land, three out of four El Niño-dominated periods (2002-2005, 2014-2016, and 2018-2020)
consistently showed negative SMB (Fig. 4e, g–h) and ice mass anomalies (Fig. 4i, k–l). The negative anomaly
signal during the 2014-2016 El Niño-dominated period is weaker compared to the 2002–2005 and 2018–2020
periods, with a weak positive anomaly observed in western Dronning Maud Land. In contrast, the negative
anomalies during the 2002–2005 and 2018–2020 periods were more widespread across Dronning Maud Land,
with slightly stronger signals in the western areas.
During the 2002–2005 El Niño-dominated period, a low-pressure anomaly over the Atlantic extending into
Dronning Maud Land, combined with a high-pressure anomaly over the continent, produced southerly and
southeasterly winds in Dronning Maud Land (Fig. 4a). Similarly, during 2018–2020, slightly weaker high-
pressure anomalies over Antarctica induced southerly wind flow off Dronning Maud Land (Fig. 4d). In contrast,
during 2014–2016, a low-pressure anomaly off the Dronning Maud Land coast generated northerly winds into
western regions—supporting slight positive anomalies—while southerly winds influenced eastern regions,
creating differential impacts (Fig. 4c, g, k).





The 2009–2010 El Niño-dominated period exhibits a markedly different response in Dronning Maud Land compared to the generally negative mass anomalies observed during other periods. Instead of negative anomalies, 2009–2010 is characterised by positive mass anomalies (Fig. 4f, j), particularly in eastern Dronning Maud Land, as shown in GRACE data (Fig. 4j). A mid-latitude blocking pattern, with a high-pressure anomaly extending as a ridge to the Antarctic coastline, drives northerly winds onshore (Fig. 4b).

In Wilkes Land, two distinct response patterns emerge across the four El Niño-dominated periods: 2002–2005 and 2009–2010 coincided with positive SMB (Fig. 4e) and ice mass anomalies (Fig. 4i), while 2014–2016 and 2018–2020 correspond to negative anomalies (Fig. 4g–h, k–l). High-pressure anomaly over Antarctica during 2002–2005 and 2009–2010 (Fig. 4a, c) align with the negative SAM phase (Fig. 1e), characterised by weakened mid-latitude westerlies and expanded high pressure over Antarctica (Marshall, 2003), which extends northward over Wilkes Land, with circulation patterns inducing northeasterly wind anomalies along the coast (Fig. 4a, c).

The 2014–2016 El Niño-dominated period aligns with low-pressure anomaly over Antarctica and intensified mid-latitude westerlies (Fig. 4c). The low-pressure anomaly located over Wilkes Land produces southerly to southwesterly wind anomalies (Fig. 4c), negatively impacting mass balance (Fig. 4g, k). During 2018–2020, weak pressure anomalies over the continent near Wilkes Land accompanied a developing low-pressure system in the adjacent ocean (Fig. 4d), intensifying offshore southerly winds and further negatively influencing mass balance (Fig. 4h, l).

### 3.3. La Niña-dominated periods

Figure 5 presents atmospheric circulation patterns, SMB anomalies, and AIS mass changes during La Niña-dominated periods. Instrument malfunctions and the termination of the GRACE mission in 2017 introduced noise and data gaps, affecting ice mass estimates. Therefore, we limit our discussion to the atmospheric circulation and SMB for the 2016–2018 La Niña-dominated period to avoid conclusions based on potentially unreliable data in GRACE.





**Figure 5. Composite anomaly maps of sea level pressure and 10 m wind from ERA5 reanalysis, representing mean anomaly conditions during La Niña-dominated periods relative to the climatology of the GRACE period (2002–2022). Sea level pressure anomalies are shown as shaded regions (hPa), while wind anomalies are indicated by reference vectors (m s⁻¹). The rate of change in Antarctic SMB is derived from the RACMO2.3p2 model (kg m⁻² y⁻¹), and the rate of surface mass change is obtained from GRACE data (kg m⁻² y⁻¹). SMB and GRACE maps illustrate trends in ice mass variability for each identified La Niña-dominated period. The GRACE signal is strongest near the coastal regions and weaker in the interior, where uncertainties are higher. The GRACE satellite malfunction during 2016-2018 is apparent in the signal for that period, where instrument noise dominates over actual variability with pronounce north-south striping.**

### 3.3.1. Atmospheric circulation and mass anomalies in West Antarctica

La Niña-dominated periods are characterised by a low-pressure anomaly in the Pacific sector, reflecting a strengthening and/or shift of the ASL (Fig. 5a–d). Our analysis reveals variable mass changes in West Antarctica between the Amundsen sector and Antarctic Peninsula, with a notable exception during 2010–2014, when a uniformly negative response was observed (Fig. 5f, j). During this period, the low-pressure anomaly in the Pacific sector extended from the Bellingshausen to Amundsen Seas (Fig. 5b). In contrast, other La Niña-dominated periods exhibited a bipolar mass pattern between the Amundsen sector and Antarctic Peninsula (Fig. 5e, g–h), with a less elongated low-pressure anomaly in the Pacific (Fig. 5a, c–d).

The low-pressure anomaly in the Pacific during these La Niña-dominated periods enhanced southerly wind anomalies off the Amundsen Embayment, with broadly negative SMB (Fig. 5f, g–h) and ice mass anomalies (Fig. 5j, k–l) consistent across the 2010–2014, 2016–2018, and 2020–2022 La Niña-dominated periods. The 2007–2009 La Niña-dominated period, however, showed a broadly positive ice mass anomaly in the Amundsen Embayment (Fig. 5e, i)—more typical of most El Niño-dominated periods as described previously—due to a northwest shift of the low-pressure anomaly in the Pacific (compared to the other three La Niña-dominated periods) (Fig. 5a). Around the Bellingshausen and Amundsen Seas, there is an interaction between northerly winds from the Pacific and southerly winds from the continent that potentially can support convection and positive mass anomalies (Fig. 5a).

The spatial impact of the Antarctic Peninsula mass responses during La Niña-dominated periods also exhibits variation, with both positive and negative mass anomalies observed across different La Niña-dominated periods. The 2007–2009 and 2010–2014 La Niña-dominated periods showed negative mass anomalies (Fig. 5e–f, i–j), while 2016–2018 and 2020–2022 La Niña-dominated periods exhibited positive anomalies (Fig. 5g–h, k–l). The widespread mass reduction during the 2010–2014 La Niña-dominated period, evident in both SMB and GRACE data (Fig. 5f, j), coincided with the strongest La Niña event in the GRACE record. Southerly winds prevailed across the Peninsula during mass loss periods (2007–2009, 2010–2014) (Fig. 5a–b), whereas northerly winds dominated during mass gain periods (2016–2018, 2020–2022) (Fig. 5g–h).



The 2020–2022 La Niña-dominated period stands out as the only one coinciding with a positive SAM phase (Fig. 1d, e), featuring an anomalous deepening of the low-pressure anomaly in the Pacific ASL. This intensified low-pressure anomaly drove strong northerly wind anomalies over the Antarctic Peninsula (Fig. 5d).

### 3.3.2. Atmospheric circulation and mass anomalies in East Antarctica

The East Antarctic coastline experienced widespread positive SMB anomalies during the 2010–2014 and 2020–2022 La Niña-dominated periods (Fig. 5f, h), while the 2007–2009 and 2016–2018 La Niña-dominated periods showed regionally variable responses across the Atlantic and Indian Ocean sectors (Fig. 5e, g). In Dronning Maud Land, SMB (Fig. 5f, h) and GRACE-derived (Fig. 5j, l) mass anomalies were consistently positive during 2010–2014 and 2020–2022 La Niña-dominated periods, whereas 2007–2009 La Niña-dominated period showed contrasting responses—positive mass anomalies in the west and negative mass anomalies in the east (Fig. 5e).

These varying impacts in Dronning Maud Land stem from the positioning of positive pressure anomaly in the Atlantic Ocean. During 2007–2009, a high-pressure anomaly west of Dronning Maud Land flow, generating northerly winds in the west and southerly winds offshore in the east (Fig. 5a), creating spatial heterogeneity in mass change (Fig. 5e). In 2010–2014, the high-pressure anomaly was farther north (Fig. 5b), resulting in uniform northerly winds and positive mass anomalies across the region (Fig. 5f). The 2020–2022 period, marked by an anomalously deep low-pressure anomaly in the Pacific, also featured strong northerly winds over Dronning Maud Land (Fig. 5d).

Wilkes Land exhibited two distinct SMB responses across La Niña-dominated periods. Positive SMB anomalies occurred in 2007–2009 and 2020–2022 (Fig. 5e, h), while 2010–2014 and 2016–2018 were associated with negative SMB anomalies (Fig. 5j, l)). The 2007–2009 period featured ridging of the Antarctic high, inducing northerly wind anomalies that support moisture transport into the region (Fig. 5a). In contrast, during 2010–2014, a weaker low-pressure anomaly generated southerly wind anomalies (Fig. 5b), likely suppressing moisture transport and leading to negative mass anomalies (Fig. 5f, l).

The 2020–2022 La Niña-dominated period is distinct due to the anomalous deepening of the low-pressure anomaly in the Pacific, which induced strong northerly-to-northwesterly winds over Wilkes Land (Fig. 5d), contributing to a pronounced positive mass anomaly (Fig. 5h, l). However, this period also includes extreme events such as the March 2022 atmospheric river event, which delivered record-breaking precipitation (Wille et al., 2024; Wille et al., 2022; Wille et al., 2021). This raises the question of whether the combination of La Niña and positive SAM increases the likelihood of such extremes, while also considering the potential impact of climate change. However, we note that there is only one example of these conditions co-occurring during the GRACE observation period.

Figure 6 presents the mean AIS response across El Niño- and La Niña-dominated periods, summarizing the impacts of different ENSO events. While this mean response differs slightly from the regression results in Fig. 3b–c, certain regional patterns remain consistent. From the GRACE results, it is obvious north-south striping noise in GRACE observations is maximised over short periods. SMB results show a positive response in the Amundsen Sea sector during El Niño-dominated periods and a negative response during La Niña-dominated periods, with the opposite pattern in the Antarctic Peninsula and Dronning Maud Land.





However, averaging multiple ENSO-dominated periods may obscure variability and lead to misinterpretation. As
shown in Figs. 4e–h and 5e–h, mass variability—particularly in the Antarctic Peninsula and East Antarctica—
varies significantly across individual ENSO events. The mean response fails to capture these short-term variations,
which are critical for understanding their influence on AIS mass balance.

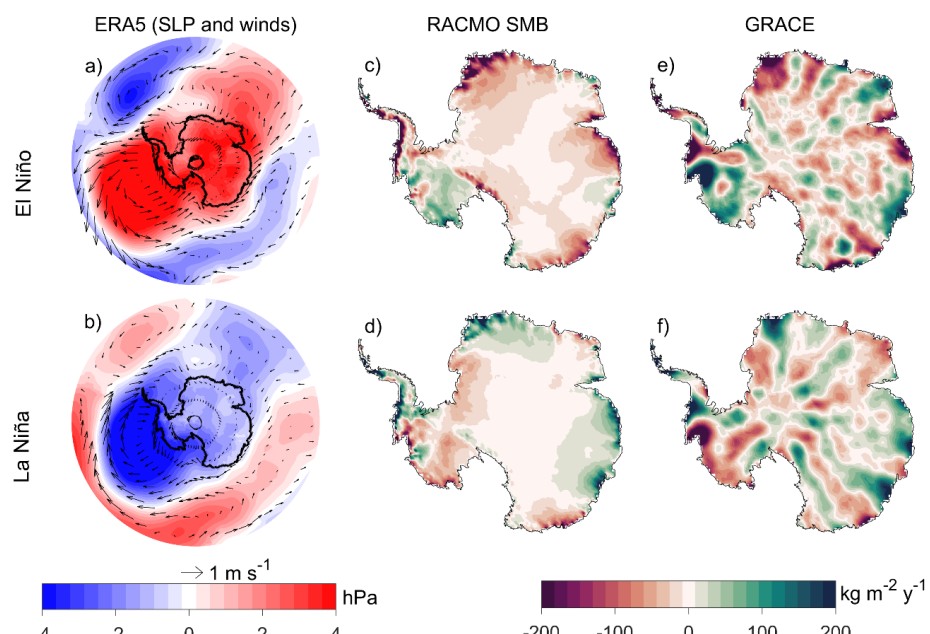

**Figure 6. The integrated spatial anomalies of climatic variables, RACMO SMB, and GRACE-derived ice**
**mass change for the four El Niño- and La Niña-dominated periods. This represents the cumulative impact**
**of different ENSO phases on AIS mass variability.**
**4. Discussion**
**4.1 Continental-wide perspective**
AIS mass variability in response to ENSO forcing is complex, as it impacts southern high latitude atmospheric
circulation, which in turn influences precipitation and Antarctic SMB (King et al., 2023; Li et al., 2022a). Our
results are consistent with previous studies, indicating that changes in atmospheric circulation linked to both
ENSO and SAM drives the short-term AIS mass variability (Clem et al., 2016; Zhan et al., 2021; Zhang et al.,
2021). The El Niño- and La Niña-dominated periods we examined exhibited consistent spatial patterns of ice mass
change during most periods across Antarctica, with mass gain in West Antarctica and mass decrease in East
Antarctica during El Niño-dominated periods and the reverse pattern during La Niña-dominated periods. This
pattern has been identified in previous studies using GRACE (King et al., 2023; Zhang et al., 2021).
However, our results reveal varied spatial patterns during different ENSO-dominated periods, not all consistent
with the previously reported bipolar ENSO spatial pattern (Li et al., 2022a; King et al., 2023). This bipolar spatial



pattern likely represents the underlying ENSO impact on the AIS, supported by the published purely data-driven
analyses of GRACE data, showing strong correlation with cumulative ENSO indices (King et al., 2023). Our
analysis suggests significant deviations from this pattern during some ENSO-dominated periods.
The interaction between ENSO strength, duration, SAM phase, and the ASL response is crucial in determining
AIS mass variability. These factors influence atmospheric circulation patterns and subsequently affect ice mass
variability across different Antarctic regions. It is noteworthy that the period from 2000 to 2020 has been
characterised by unusual Antarctic climate dynamics, attributed to changes in large-scale circulation patterns that
have significantly altered climate patterns across the continent (Xin et al., 2023).
**4.2 West Antarctica**
**4.2.1 El Niño-dominated periods**
The spatial SMB and mass change patterns we observed across the West Antarctic Ice Sheet largely correspond
to the position and intensity of the ASL (Raphael et al., 2016a). The 2002-2005, 2014-2016 and 2018-2020 El
Niño-dominated periods showed mass gain patterns, especially in the Amundsen Sea region (Fig. 4e, g–h),
consistent with previous studies (Paolo et al., 2018; King et al., 2023). This positive mass anomaly pattern is
supported by the atmospheric circulation during these periods (Fig. 4a, c–d), as ASL variability modulates
moisture transport into the Amundsen Sea sector and the Antarctic Peninsula (Raphael et al., 2016a).
During El Niño conditions, the weakening of the ASL and coastal easterlies reduces Ekman transport of cold
surface water onto the continental shelf, enhances on-shelf transport of warm Circumpolar Deep Water, and leads
to subsurface warming of the continental shelf (Paolo et al., 2018; Huguenin et al., 2024). Despite increased basal
melting during El Niño events, high snow accumulation from moisture-laden winds undergoing orographic lifting
offsets mass loss at the surface, contributing to a positive mass anomaly (Huguenin et al., 2024). Our results
support these findings, as three out of four El Niño-dominated periods show a positive anomaly in the Amundsen
Sea sector, as seen in both SMB and GRACE.
The mass gain in these 2002-2005, 2014-2016 and 2018-2020 El Niño-dominated period in the Amundsen sector
differ from findings reported by Macha et al. (2024), who noted an increase in snow accumulation in the western
Ross Sea sector and a decrease in the Amundsen Sea sector for both central and eastern Pacific El Niño events.
This discrepancy stems from methodological differences: our analysis focuses on the net mass change over entire
ENSO-dominated periods (defined using the Niño3.4 index), whereas Macha et al. (2024) utilised the central and
eastern El Niño indices provided by Ren and Jin (2011) and concentrated on seasonal mass changes during the
peak (JJA and SON) of these ENSO events. By considering entire periods rather than just peak phases, our
approach captures the net mass change throughout complete events, providing important context for overall ice
sheet mass balance.
Our results from the 2009-2010 El Niño-dominated period show a pattern across the atmospheric circulation,
SMB and GRACE that is more typical of La Niña events (Fig. 4b, f, j), with offshore winds and a decrease in ice
mass observed in West Antarctica. This is unexpected since El Niño events typically enhance moisture transport
into West Antarctica (Huguenin et al., 2024). This period has been characterised as a strong Central Pacific El



Niño (Kim et al., 2011), and the anomalous response can be attributed to altered Rossby wave propagation from
the tropical Pacific into Antarctica (Chen et al., 2023). This shift generates an anticyclone in the Ross-Amundsen
seas, inducing southerly flow, which reduces precipitation between the Amundsen and Bellingshausen regions
(Chen et al., 2023). Our SMB pattern for the 2009-2010 El Niño-dominated period, particularly in West
Antarctica, aligns closely with the influence of Central Pacific El Niño events on Antarctic SMB as described by
Macha et al. (2024).
The Antarctic Peninsula's response to El Niño-dominated periods shows considerable variability in SMB and ice
mass changes. This variability is closely linked to large-scale climate modes, such as SAM and ENSO (Clem et
al., 2016), as well as the Peninsula's unique geography, which is marked with by the presence of a mountain range.
SAM particularly affects westerly winds and associated moisture delivery to the Peninsula, especially its western
side (Orr et al., 2008).
During the 2002-2005 and 2014-2016 El Niño-dominated periods (Fig. 4a, c), westerly winds transported moisture
onto the western side of the Peninsula, leading to increased precipitation through orographic lifting and resulting
in positive SMB and ice mass anomalies. The westerlies were particularly strong during the 2014-2016 period
due to the prevailing positive phase of SAM, which enhanced moisture transport and contributed to a stronger
positive SMB anomaly compared to the 2002-2005 period.
On the eastern side of the Peninsula, westerlies typically induce foehn winds (Clem et al., 2016; Clem and Fogt,
2013), reducing SMB and ice mass. This pattern was present during both the 2002-2005 and 2014-2016 El Niño-
dominated periods, though the signal remained relatively weak (Fig. 4e, g). In contrast, the sustained influx of
moisture from the deepened ASL during the 2020–2022 La Niña-dominated period favoured precipitation, leading
to positive SMB anomalies and net ice mass gain.

### 4.2.2 La Niña-dominated periods

La Niña-dominated periods showed varied effects (Fig. 5). Two out of three La Niña-dominated periods we
considered (excluding the noisy GRACE solution during the 2016-2018 La Niña-dominated period) display a
consistent spatial pattern, with negative mass anomalies in the Amundsen Sea region (Fig. 5j, l), aligning with
previous studies (Paolo et al., 2018; King et al., 2023; King and Christoffersen, 2024). The strengthening of the
ASL during La Niña conditions enhances coastal easterly anomalies in West Antarctica (Fig. 5b, d), increases
Ekman transport of cold surface water onto the ice shelf, and reduces the on-shelf transport of warm Circumpolar
Deep Water and moisture-laden winds, leading to reduced precipitation in West Antarctica (Huguenin et al.,
2024). The intensification of the ASL during La Niña events inhibits moisture influx into the region by promoting
offshore winds  (Hosking et al., 2013), resulting in reduced precipitation, SMB, and ice mass decline in West
Antarctica (King et al., 2023; Zhang et al., 2021). However, a positive mass anomaly occurs in the Amundsen Sea
sector during the 2007-2009 La Niña-dominated period, contrasting with other La Niña-dominated periods (Fig.
5i). This highlights the fact that ENSO is not the sole driver of ice mass variability in West Antarctica, though our
analysis is limited in isolating ENSO signals in the region. The positive mass anomaly may potentially be tie to
the interaction between the northerly and southerly winds, which form a convergence zone that enhances
precipitation (Fig. 5a).



Ice mass variability in the Antarctic Peninsula is complex, as ENSO and SAM influence on circulation patterns
differs spatially and seasonally in terms of temperature variability (Clem et al., 2016; Clem and Fogt, 2013).
ENSO conditions tend to promote meridional circulation, especially during winter, while SAM favours zonal
circulation. Together, these create complex effects on Antarctic Peninsula climate. Studies report decreased SMB
along the Bellingshausen Sea-Antarctic Peninsula during El Niño, with an increase in the Amundsen Sea sector,
while the reverse occurs during La Niña (Sasgen et al., 2010). This spatial pattern is consistent with our results
for the 2018-2020 El Niño- and 2020-2022 La Niña-dominated periods (Figs. 4l and 5l). Other studies have
reported a uniform impact spanning from the Amundsen Sea sector to the Antarctic Peninsula (Zhang et al., 2021),
which aligns with our observed ice mass change during the 2014-2016 El Niño and 2010-2014 La Niña-dominated
periods (Figs. 4g and 5j). The impact of ENSO between the Amundsen Sea sector and Antarctic Peninsula depends
on the location and extent of the ASL between the Ross and Bellingshausen Seas (Raphael et al., 2016b). During
La Niña, the ASL tends to elongate, with its centre often located further west compared to its position during El
Niño (Huguenin et al., 2024), and a boarder ASL leads to more uniform impact across West Antarctica (Clem and
Fogt, 2013), as observed during the 2010-2014 La Niña-dominated periods.
**4.3 East Antarctica**
In East Antarctica, moisture transport appears primarily influenced by the strength and position of cyclonic and
anticyclonic anomalies over the continent and the Southern Ocean (Figs. 4a–d and 5a–d). These pressure
anomalies regulate atmospheric circulation, with changes in the meridional atmospheric flow affecting heat and
moisture distribution across the region (Scarchilli et al., 2011; Wang et al., 2024; Udy et al., 2021). The pressure
anomaly over the Antarctic continent is largely governed by the SAM phase, which modulates the positioning of
cyclonic and anticyclonic anomalies over both the continent and Southern Ocean, establishing SAM as a key
driver of East Antarctic Ice Sheet variability.
**4.3.1 El Niño-dominated periods**
ENSO impacts West Antarctica through modulation of the ASL via Rossby wave propagation, though the ASL's
influence on East Antarctica remains unclear. ENSO-induced pressure anomalies in the Pacific Ocean can
potentially influence moisture inflow into East Antarctica through the ASL (Li et al., 2022a), as observed during
the 2020-2022 La Niña-dominated period (Fig. 5a). During El Niño-dominated periods, the weakening of the ASL
in three out of the four El Niño dominated periods aligns with the formation of a low-pressure anomaly in the
South Atlantic (Fig. 4a, c–d). This South Atlantic low-pressure anomaly, previously associated with El Niño
events (Li et al., 2022a), induces equatorward wind flow (cold and dry southerly anomalies), leading to decreased
precipitation, reduced SMB and negative mass anomalies in the Atlantic sector of East Antarctica (Fig. 4e, g–h).
In contrast, the Atlantic sector experienced mass increase during the 2009-2010 El Niño-dominated period. The
significant mass gain observed in Dronning Maud Land (Atlantic sector) during this period has been attributed to
atmospheric blocking, which produced episodic snowfall events (Boening et al., 2012). Atmospheric blocking
favours the occurrence of atmospheric rivers reaching the Antarctic coastline and is often associated with
increased precipitation and temperature (Wille et al., 2021). The weakening of the westerlies during negative
SAM conditions (Clem et al., 2016), allows for Rossby wave amplification and an increased frequency of
atmospheric blocking events in East Antarctica, particularly during winter, when the relationship is strongest



(Wang et al., 2024). However, no statistically significant relationship has been established between negative SAM
and atmospheric river frequency in Dronning Maud Land (Wille et al., 2021) These blocking events significant
impact East Antarctic climate, through their influence on temperatures and precipitation (Wang et al., 2024; Udy
et al., 2021; Pohl et al., 2021).
The 2014-2016 El Niño-dominated period demonstrated a spatial pattern in East Antarctica that closely aligned
with a positive SAM signal response, resulting in a negative mass anomaly in the region (Fig. 4g). During this
period, the strengthened westerlies around 60°S, associated with positive SAM, enhanced moisture transport away
from Antarctica, reducing precipitation and leading to the observed negative mass anomaly (Marshall et al., 2017).
However, differentiating the timescale between individual extreme snowfall events and ice sheet response in
monthly GRACE-observed ice mass data is complex. Atmospheric rivers, for instance, occur on average less than
5 days per year but can contribute 30-40% of annual precipitation (Wille et al., 2021). Despite the short duration
of these events, the impact of ENSO on SMB can be influenced by synoptic scale phenomena, such as atmospheric
rivers associated with blocking events (Pohl et al., 2021). These high-impact, short-term events can disrupt
expected ENSO patterns, leading to varied impacts on the ice sheet, as observed in the positive mass anomaly in
Dronning Maud Land during the 2009-2010 El Niño-dominated period.
**4.3.2 La Niña-dominated periods**
La Niña has been linked to high-pressure anomaly development in the South Atlantic, which leads to moisture
advection into Dronning Maud Land (Li et al., 2022b). This moisture transport results in increased precipitation
and a subsequent positive mass anomaly in the region. In two out of the three La Niña-dominated periods (2007-
2009, 2010-2014 periods) considered; a high-pressure anomaly in the South Atlantic is a common feature (Fig.
5a–b).
Similar to El Niño-dominated periods, the response of the East Antarctic Ice sheet during La Niña-dominated
periods also shows variability, with both consistent and opposing anomaly signals between the Indian and Atlantic
sectors. During the 2010-2014 La Niña-dominated period, Dronning Maud Land experienced a positive mass
anomaly, while Wilkes Land showed a negative mass anomaly (Fig. 5f). By contrast, for the 2020-2022 La Niña-
dominated period, both Dronning Maud Land and Wilkes Land exhibited a positive mass anomaly (Fig. 5h),
suggesting that, in addition to high-pressure anomalies driving moisture into the region, other factors also
influence these regional responses.
We observed large mass gain during the 2020-2022 La Niña-dominated period (Fig. 5h); however, this gain cannot
be directly attributed to the amplification of positive SAM and La Niña anomalies, as they appear to be
atmospherically in phase (Fig. 1c). Our analysis does not account for the removal of the extreme March 2022
heatwave event, which saw record-shattering temperature anomalies and widespread snow accumulation (Wille
et al., 2024). However, the deepened low-pressure anomaly in the Pacific induced strong northerly winds across
the Peninsula into Dronning Maud Land, while the symmetric structure of the westerlies was altered, allowing
northerly winds to reach Wilkes Land.





Ice sheet variability in the Indian sector is influenced by multiple factors and not solely driven by ENSO signals.
SAM signals have been found in Wilkes Land (King and Christoffersen, 2024; King et al., 2023), and synoptic
weather patterns in the southern Indian Ocean can influence the transport of moisture and heat into the region,
ultimately affecting ice mass variability (Udy et al., 2021).
Our findings indicate that ice mass changes during ENSO-dominated periods cannot be solely attributed to ENSO
forcing. To quantify changes in ENSO variability, long time series must be considered in future studies (Stevenson
et al., 2010), along with the use of climate models to better isolate and capture purely ENSO-driven signals.
**4.4 Combined ENSO and SAM influence**
Isolating the ENSO signal and its impact on AIS ice mass is challenging due to several factors. The Rossby wave
propagation of the ENSO signal to Antarctica is influenced by SAM (Marshall, 2003; Fogt and Marshall, 2020),
and the ENSO signal can be masked by other climate modes, such as zonal-wave 3—a quasi-stationary pattern in
the southern high latitudes that affects meridional heat and momentum transport (Goyal et al., 2022; Raphael,
2004). Additionally, synoptic-scale weather systems can further mask ENSO's influence. The complex interaction
between ENSO and other modes of climate variability likely drives the equally complex patterns of AIS ice mass
change observed during different ENSO-dominated periods.
While our analysis does not explicitly resolve the mechanisms through which ENSO and SAM influence wind
anomalies, previous studies have demonstrated strong correlations between ENSO and meridional winds, and
between SAM and zonal winds, both significantly influencing Antarctic Peninsula climate (Clem et al., 2016).
Our analysis, which uses cumulative summed indices to match GRACE mass time series, has limitations. It
focuses primarily on low-frequency variability and does not account for shorter temporal scale impacts, such as
tropical convection pulses that trigger the Rossby waves or high-frequency variability associated with storm
systems such as atmospheric rivers. However,the net effect of these would be captured by GRACE.
The combination of La Niña and positive SAM conditions strengthen the ASL (Fogt et al., 2011) and drives
positive temperature anomalies across the Antarctic Peninsula and East Antarctica (Clem and Raphael, 2023).
This relationship partially explains the significant mass gain observed across these regions during the 2020-2022
La Niña-dominated periods. The extreme atmospheric river event in March 2022 largely contributed to the
observed mass gain over this ENSO-dominated period (Wille et al., 2024; Wille et al., 2022).
Studies on precipitation (Marshall et al., 2017) and ice core records (Medley and Thomas, 2019) both recognise
that SMB generally decreases during positive SAM phase and increases during negative SAM phase. In terms of
the impact on basal melting, negative SAM periods generally decrease the transport of warm circumpolar deep
water onto the continental shelf (Palóczy et al., 2018), largely reducing ice shelf basal melt (Verfaillie et al., 2022)
and subsequently contributing to ice mass gain. However, the timescale of the response of the upstream ice to the
positive SAM forcing is unclear and  would involve a substantial lag (King and Christoffersen, 2024). The spatial
pattern of ice mass change anomaly during the 2002-2005 El Niño and 2007-2009 La Niña-dominated periods in
the Amundsen Sea sector and Wilkes Land resembles the negative SAM spatial pattern reported by King et al.



(2023). Negative SAM dominates the cumulative summed SAM (Fig. 1e) from the start of the GRACE time series
in 2002 until around 2008, which aligns with the high-pressure anomaly observed over Antarctica, reflecting a
stronger than average (over the GRACE period) Antarctic high during this period (Figs. 4a–b and 5a). Therefore,
it is possible that ice mass variability observed between 2002 and 2008 was more influenced by SAM than by
ENSO.
Our findings agree with an understanding that ENSO forcing on the Antarctic climate impacts atmospheric
circulation patterns, altering the ASL variability, which in turn influences Antarctic ice mass variability (Zhang
et al., 2021; Paolo et al., 2018; Sasgen et al., 2010; Clem et al., 2017). The internal dynamics of the ASL may
contribute to AIS mass variability that is independent of the influence of ENSO and SAM which potentially can
impact our analysis. Given that our analysis spans a 22-year period, these results primarily capture the interannual
variability rather than lower frequency influence of the Pacific Decadal Oscillation signal. While ENSO induced
circulation affects Antarctic SMB (Kim et al., 2020), recent Antarctic ice mass trends (2003-2020) have been
primarily driven by mass imbalance triggered by long-term ice dynamics changes (Kim et al., 2024; Rignot et al.,
2019). Some of the low-frequency mass variability around the long-term trend (which we remove) is associated
with changing ice dynamics. This dynamic signal is stronger in West than in East Antarctica.
In a warming climate, future ENSO event variability is predicted to increase (Cai et al., 2021). CMIP5 model
simulations suggest a reduction in El Niño-induced precipitation over West Antarctica (Lee et al., 2023). Given
that SAM is projected to remain in its positive phase across all seasons due to greenhouse gas emissions (Arblaster
and Meehl, 2006), accurate modelling of future AIS mass estimates in relation to ENSO teleconnections must
account for the interaction between SAM and ENSO. The AIS mass gain observed during 2020-2022 raises
questions about how the AIS will respond to future La Niña and positive SAM periods and if it would increase
the frequency of extreme events.
**5 Conclusion**
To examine the AIS mass change during different ENSO-dominated periods, we analysed AIS mass change
anomalies observed by GRACE/GRACE-FO. These anomalies were interpreted alongside modelled SMB and
atmospheric pressure and wind patterns. Our analysis reveals that El Niño and La Niña exert distinct influences
on AIS, with considerable event-to event variability.
At the continental scale, three out of the four El Niño-dominated periods were characterised by mass increase in
West Antarctica and mass decrease in East Antarctica. Conversely, two out of the three La Niña-dominated
periods (here excluding the 2016-2018 period with degraded GRACE signal) showed the opposite pattern, with
mass reduction in West Antarctica and mass increase in East Antarctica. The Amundsen Sea sector typically
experiences positive mass anomalies during El Niño-dominated periods and negative anomalies during La Niña-
dominated periods. In East Antarctica, a consistent mass increase was observed during two out of three La Niña-
dominated periods.
Mass variability in West Antarctica is primarily driven by ENSO-induced ASL pressure anomalies, which
modulate the atmospheric circulation and moisture transport. The ASL exhibits high variability in its location,



strength, and extent, which influence its impact between the Antarctic Peninsula and West Antarctica. The ASL
strengthens and moves closer to the Antarctic coastline during periods when ENSO-SAM are in phase  (Hosking
et al., 2013). While ENSO has its strongest impact in West Antarctica, its influence extends to East Antarctica,
consistent with Li et al. (2022a). However, atmospheric pressure patterns over the Southern Ocean play a crucial
role in regulating moisture influx and, consequently, ice mass variability in East Antarctica.
In summary, this study highlights the complex nature of ENSO teleconnections in modulating AIS mass balance
through changes in atmospheric circulation. Rather than exhibiting a simple dipole response (Fig. 3), AIS mass
variability during ENSO events is shaped by unique teleconnections and moisture fluxes specific to each event.
Although climate model projections remain uncertain regarding whether future ENSO events will more resemble
an El Niño- or La Niña-like state, they consistently indicate that ENSO will influence Antarctic precipitation
patterns. A clearer understanding of ENSO's role in Antarctic climate is therefore critical for assessing its impact
on future SMB and long-term ice mass balance. This requires both process-level understanding (e.g., Macha et
al., 2024) and consideration of the net effect on ice sheet mass as explored here.
**Code and Data availability**
Source code and data will be made available through the University of Tasmania Research Data Portal prior to
publication. The GRACE data used is available at https://gravis.gfz.de/ais. The ERA5 reanalysis data used in the
atmospheric linkage to ice mass variation are publicly available from https://cds.climate.copernicus.eu/. The
station-derived SAM index from  Marshall (2003) available at http://www.nerc-bas.ac.uk/icd/gjma/sam.html.
The Niño3.4 index are publicly available from https://psl.noaa.gov/data/timeseries/month/Nino34/.
RACMO2.3p2 model SMB output can be accessed at (Van Dalum et al., 2021).
**Author contributions**
All authors contributed to the conception and design of the study. JBA performed the statistical analysis and data
processing. JBA wrote the manuscript with input from all co-authors. All authors helped with the revision and
approved the final version of the manuscript.
**Competing interests**
The authors declare that they have no conflict of interest.
**Disclaimer**
Publisher's note: Copernicus Publications remains neutral with regard to jurisdictional claims made in the text,
published maps, institutional affiliations, or any other geographical representation in this paper. While Copernicus
Publications makes every effort to include appropriate place names, the final responsibility lies with the authors.



**Acknowledgements**
We thank the GravIS team for supplying GRACE data, the European Centre for Medium-Range Weather
Forecasts for providing reanalysis climatic data, NOAA for the ENSO indices, Marshall (2003) for the SAM index
and finally Van Wessem (2023) for providing the SMB dataset.
**Financial support**
JBA, MK and DU were supported by the Australian Research Council Special Research Initiative, Australian
Centre for Excellence in Antarctic Science (Project Number SR200100008). TV was supported by the Australian
Government's Antarctic Science Collaboration Initiative (ASCI000002) through funding to the Australian
Antarctic Program Partnership. JBA was supported by a University of Tasmania Graduate Research Scholarship.

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
