# Peer review of "The changing mass of the Antarctic Ice Sheet during ENSO- 2 dominated periods in the GRACE era (2002–2022)"

_EGUsphere, 2025_

## Author Comment (AC1)

**Reviewer 1**

**SUMMARY**

The study presents the impact that different ENSO-induced atmospheric circulation changes have on Antarctic ice sheet mass changes and analyze teleconnections with the southern annular mode. The authors show that there is strong event-to-event spatial variability between ENSO events using GRACE observed mass changes, regional climate model output and ERA5. This work fits well within the scope of the journal and provides a contribution to the field. The manuscript is generally well written, but some paragraphs can be somewhat lengthy. The following comments should help with solving the remaining issues before publication, with e.g. L1 referring to line 1.

**Author's response**:   We thank you for your thoughtful and constructive comments, which have contributed to improving the clarity, structure, and scientific integrity of the manuscript. All major and minor points have been carefully addressed.

**General comments:**

**Reviewer comment**

- Recently, a new version of the regional climate model RACMO2.4p1 was published for the Antarctic ice sheet (Van Dalum et al., 2025, https://doi.org/10.5194/egusphere-2024-3728), which includes new physics (in particular relevant here are changes in precipitation). Importantly, RACMO2.4p1 also has a higher horizontal resolution of 11 km compared to the 27 km resolution used in RACMO2.3p2. Using the SMB of RACMO2.4p1 would improve the comparisons done in this study and I suggest the authors to use this version instead of RACMO2.3p2. RACMO2.4p1 data can be found here: https://doi.org/10.5281/zenodo.14217231

**Author's response**: In this study, we are primarily interested in the broad patterns associated with ENSO and the impact of the update of RACMO needs exploration. We will examine the differences produced by RACMO2.4p1 and, if they are substantial, we will update the results. If they are not, we will make note of their similarity.

**Reviewer comment**

- In the manuscript, basal melting is mentioned but SMB and mass changes are not studied on the ice shelves, hence relating the results to basal melting is difficult. Therefore, consider to include ice shelves in the comparison with RACMO SMB in e.g. Fig. 3b and elsewhere, and if possible also for GRACE, or explain why that cannot be done. Furthermore, it is also interesting to see how the SMB changes over the major ice shelves for each ENSO period.

**Reviewer comment**

**Author's response**:   We did not examine basal melting in this study. However, as GRACE data cannot distinguish between mass changes due to atmospheric forcing and those due to ice dynamics, our mention of basal melting refers to components of ice mass change that are potentially not explained

by atmospheric processes. We will revise this section of the manuscript for better clarity and to avoid any confusion.

**Reviewer comment**

- I think it is valuable for this study to mention whether an ENSO event is central or eastern and discuss if and how such events differ, as it may explain some of the patterns that are identified in this study and therefore increase understanding. The authors shortly discuss the potential importance in the manuscript, like on L486-495, but I think a more in-depth analysis will improve the manuscript. Other work, like Macha et al. (2024), may provide information about whether an ENSO event is central or eastern, or it can be determined by following methods described by Ren and Jin (2011).

**Author's response**: The use of cumulatively summed ENSO indices allows us to capture the net influence of all ENSO events within a period, including transitions between central and eastern Pacific events. This means that our ENSO periods may cover a single ENSO event, or they may cover a series of events, such as two or three La Niña events in a 2-3 year period. Because of this, it is a different technique to that of Macha et al., as one of our 'ENSO events' may cover both central and eastern Pacific events. In addition the Niño3.4 index does not distinguish between central and eastern-type ENSO events, however we acknowledge the value of more detailed classification. We will expand our discussion of the Macha et al work. We agree that it would be helpful to indicate whether the El Niño–dominated periods included Central or Eastern Pacific events. Rather than assigning events on a month-by-month basis, we will refer to established classifications in the literature to identify which periods include Central or Eastern Pacific El Niño events. This will provide useful context without implying monthly resolution that our data do not support.

**Reviewer comment**

- Not all locations that are discussed in the manuscript are shown on a map, like the Wedell Sea, Ross Sea, location of the ASL or the various ocean sectors. Including the locations mentioned in the manuscript will improve clarity, making it easier to follow.

**Author's response**: As suggested, we will revise the manuscript to include more regional delineation on the maps, improving clarity and ease of interpretation.

**Reviewer comment**

- Including maps where the SMB changes are shown in percentage of the total SMB for the considered periods will help to understand how big the impact of ENSO/SAM is on the various regions that are considered, as some changes may seem large in for example high precipitation areas, while they are only relatively small. An alternative could be to report the integrated SMB values in Gt yr-1 for the ENSO events for the whole domain and smaller regions and compare them to the reference period.

**Author's response**: Since we are interested in the total mass of the ice sheet we are interested in both absolute and relative impacts. To address the relative impact, we will include maps showing SMB changes expressed as a percentage of the climatological mean SMB for each ENSO-dominated period. For each period (e.g., the 2009–2010 El Niño-dominated period), the mean SMB will be computed and compared to the long-term climatological mean at each grid point, then expressed as a percentage. These maps will highlight regions where ENSO-related atmospheric circulation changes

result in substantial deviations in SMB. However, our objective is to capture the absolute mass change rather than the relative mass change.

**Specific comments:**

**Reviewer comment:** L18: As you also use regional climate model output in your study, it should be mentioned in the abstract as well.

**Author's response:** We will include the model output in the next revised text.

**Reviewer comment:** L23-26: "… and its influence on the ASL and the Southern Ocean circulation can be equally (and in some cases more) important to AIS variability." Please specify with respect to what or rephrase this sentence.

**Author's response:** We will rephrase this sentence for better clarity.

**Reviewer comment:** Abstract: I think it is also important to shortly mention the uncertainties in the abstract that you also mention in the text, such as the relatively short time period that you use and the various teleconnections that may have not happened yet within this time period, or other processes like atmospheric rivers.

**Author's response:** We agree with the suggestion and will include it in the abstract.

**Reviewer comment:** L29-30: "The drivers of inter-annual to decadal Antarctic Ice Sheet (AIS) mass variability are complex and not yet fully understood". Please add a reference to this.

**Author's response:** Reference will be added to this statement in the revised manuscript

**Reviewer comment:** L35: Not only precipitation, but also riming can add to the SMB.

**Author's response:** We will revise the manuscript accordingly.

**Reviewer comment:** L43: Can you specify here what typically the time scale is that the SAM changes from positive to negative, or vice versa and why the SAM happens?

**Author's response:** We will elaborate on the timescale of the SAM changes and provide further explanation of the underlying mechanisms driving these variations.

**Reviewer comment:** L50: Is the total reduction of precipitation in the East AIS typically comparable to the precipitation increase in West Antarctica and the western Antarctic Peninsula? In other words, looking at the AIS as a whole, does a positive SAM increase or decrease the SMB?

**Author's response:** Overall, positive SAM phases are associated with a net reduction in SMB over the AIS, while negative SAM phases are linked to a net increase in SMB. We will revise this paragraph to more clearly reflect the overall impact of SAM on AIS SMB.

**Reviewer comment:** L67-75: Please add the location of the ASL, sectors like the Pacific sector, Indian sector etc. and other names in a map (for example in Fig. 2), which would help visualize the processes described the paper.

**Author's response:** We will include the mean locations of the ASL, as well as the Pacific, Atlantic, and Indian sectors, in Fig. 2 for clarity.

**Reviewer comment:** L76-83: Mention here why your study is different than the studies that you mention.

**Author's response:** We will clarify in the revised manuscript why this study is unique compared to previous studies, highlighting its novel contributions.

**Reviewer comment:** L87: As GRACE observes mass changes, the mass loss due to processes like runoff and sublimation are also included in the signal and should be mentioned here, even though they are relatively small compared to discharge.

**Author's response:** We will revise the manuscript as suggested.

**Reviewer comment:** L139: Please mention that the index is normalized in Fig. 1a.

**Author's response:** We will revise the manuscript as suggested.

**Reviewer comment:** L149: Also mention that the climate indices are detrended in Fig. 1c.

**Author's response:** We will revise the manuscript as suggested.

**Reviewer comment:** L155-161: Consider moving this paragraph such that it is mentioned before the paragraph of L148-154.

**Author's response:** We will revise the manuscript as suggested.

**Reviewer comment:** L162-164: "...where the positive phase of ENSO dominates the negative ENSO phase until a positive peak in the cumulative index is reached...". I think that I know what the authors mean, but consider reformulating this to improve clarity. Also, do you apply a minimum length that an ENSO period has to last?

**Author's response:** The sentence will be revised to improve clarity. We did not apply a minimum length criterion but were instead focused on the total mass change over the duration of each ENSO-dominated period.

**Reviewer comment:** Fig. 1: Please add a description to the Y-axis of the figures. In Figure 1.d, consider adding ENSO and in Figure 1.e SAM in the top of the figure, which would help reading the figure more quickly.

**Author's response:** We will revise the manuscript as suggested.

**Reviewer comment:** Section 2.3: It has not been mentioned in the paper before why you want to use a regional climate model and why it is necessary, which should be explained in e.g. the introduction before explaining what regional climate model you are going to use.

**Author's response:** A brief explanation of the rationale for using a regional climate model will be included in the revised manuscript.

**Reviewer comment:** L189: ".."at its lateral and ocean boundaries..." → at its lateral boundaries and SST and sea ice extent at the sea surface boundary...

**Author's response:** We will revise the manuscript as suggested.

**Reviewer comment:** Section 2.4: The authors should mention here why it is necessary to use ERA5 over RACMO output for the 10 m wind speeds and sea level pressure.

**Author's response**: We will provide a justification for using ERA5 instead of RACMO output for the wind and sea level pressure fields in the revised manuscript.

**Reviewer comment:** L225: Capital letter is missing in 'key'.

**Author's response**: We will revise the manuscript as suggested.

**Reviewer comment:** L227-229: Also mention here that you plot ERA5 and RACMO in Figure 3.

**Author's response**: We will revise the manuscript as suggested.

**Reviewer comment:** Fig. 3: I do not fully understand what is shown here. Is this the SLP and winds, SMB and GRACE mass loss averaged over the ENSO events? If this is the average over the ENSO events, including both El Nino and La Nina, would they not compensate each other?

**Author's response**: We will reword the methods section and corresponding figure captions to improve clarity, and we will take special care to ensure consistent and precise use of terminology throughout the manuscript. The figure shows the regression coefficient of sea level pressure (SLP) and winds anomalies (cumulatively summed), surface mass balance (SMB, cumulatively summed), and GRACE anomalies onto the cumulatively summed ENSO index. The results illustrate the atmospheric and mass patterns associated with El Niño events; conversely, the opposite pattern is generally observed during La Niña conditions. We will review the description and improve the clarity

**Reviewer comment:** Regarding Figures 4 and 5, we agree that interpreting the results would be easier with additional context on the SAM phase. We will revise the figure captions and/or figure panels to indicate whether the SAM index during each ENSO event was positive, negative, or neutral. This will help clarify how the combined phase of ENSO and SAM influences the observed spatial patterns.

**Author's response**: The manuscript will be amended accordingly.

**Reviewer comment:** Fig. 4i-l: Do you know why the north-south striping is so much more pronounced in Fig. 4j and Fig. 4l compared to Fig. 4i and Fig. 4k?

**Author's response**: It is possible that the north-south stripping is much more pronounced over shorter periods of time. Furthermore, due to instrument degradation toward the end of the GRACE mission, the observational error increases, which likely contributes to the more noticeable north–south striping in Fig. 4j and Fig. 4l.

**Reviewer comment:** L310: Do you mean Fig. 4g instead of Fig. 4c?

**Author's response**: Fig. 4c instead.

**Reviewer comment:** L311-312: "Note that the 2002-2005 SMB anomaly is only marginally positive (Fig. 4a)." → Note that the 2002-2005 SMB anomaly is only marginally positive for the Antarctic Peninsula (Fig. 4e).

**Author's response**: We will revise the manuscript as suggested.

**Reviewer comment:** L313, 314: Fig. 4f → Fig 4f, h and also Fig. 4j → Fig. 4j, l.

**Author's response**: We will revise the manuscript as suggested.

**Reviewer comment:** L323: Please also show these sectors on a map, e.g. Fig. 2.

**Author's response**: We will revise the manuscript as suggested.

**Reviewer comment:** L330-353: Link the pressure anomalies and wind changes to moisture transport and their consequent impact on SMB and mass changes. These paragraphs can also be shortened.

**Author's response:** The paragraphs will be rewritten for clarity and improved flow.

**Reviewer comment:** L380-381: Fig. 5f, g-h → Fig 5f-h and also Fig. 5j, k-l → Fig. 5j-l

**Author's response:** We will revise the manuscript as suggested.

**Reviewer comment:** L385-387: Can you explain more how the northerly winds from the Pacific and southerly winds from the continent can lead to convection? And how it may result in positive mass anomalies?

**Author's response:** We attribute the positive mass anomalies to the convergence zone formed where northerly and southerly winds meet, enhancing convection and leading to increased precipitation.

**Reviewer comment:** L393-398: Similarly, as before, link the pressure and wind anomalies to moisture transport and then to SMB and mass changes.

**Author's response:** We will revise the manuscript as suggested.

**Reviewer comment:** L421-426: How much of the 2020-2022 La Nina SMB signal is caused by this atmospheric river event? Is it possible that it is (almost) completely dominated by it?

**Author's response:** This was not included in our initial analysis, but we will incorporate an assessment to determine the extent to which the March 2022 ice mass anomaly influenced the 2020–2022 La Niña-dominated period.

**Reviewer comment:** Fig. 6: How did you calculate the average of the anomalies shown here? Did you weigh them by the length of the El Nino or La Nina-dominated periods? Or did you simply take the average of the maps that you have shown in Fig. 4 and 5?

**Author's response:** We simply averaged the anomaly maps presented in Figures 4 and 5 without weighting by period length. Our aim was to highlight the mean spatial response of ENSO. We will clarify this in the figure caption and in the results section and note it in discussion.

**Reviewer comment:** 459-461: Can you elaborate about these unusual climate dynamics? Does this have any impact on ENSO/SAM related SMB changes that you have discussed in the paper?

**Author's response:** We will expand on the unusual climate dynamics and explain how they influence the observed ENSO/SAM-related SMB variability. This will be included in the revised manuscript to provide a clearer context for the observed mass change anomalies.

**Reviewer comment:** L474-476: I am not sure if I fully understand how your results support the findings that increased basal melt is compensated by higher SMB. If I am not mistaken, you do not include ice shelves in your analysis where basal melt can occur, so how do you know that the positive SMB anomalies and increased mass that you show compensate for increased basal melt?

**Author's response:** We agree with your comment, and this statement will be omitted in the revised manuscript.

**Reviewer comment:** L477: "… El Nino-dominated period in the Amundsen sector differ" → "… El Nino-dominated periods in the Amundsen sector differs"

**Author's response**: We will revise the manuscript as suggested.

**Reviewer comment:** L483-485: As you include the complete events, doesn't it make your methods more vulnerable for irregular events, such as atmospheric rivers, that may overshadow the ENSO signals?

**Author's response**: Our methodology is vulnerable to irregular events such as atmospheric rivers (ARs). However, we can't exclude ARs from the methodology (there are multiple each year), but we can provide additional context and strengthen the discussion the likely contribution of ARs to SMB anomalies during ENSO dominating periods, using some key examples (e.g. 2009 in DML and 2022 in Wilkes Land) from the ENSO-dominated period by accounting the proportion of the signal to that AR. We will ensure that our discussion remains measured and is clearly framed within the context of Shields et al. (2022) and related work.

**Reviewer comment:** L508-510: Considering moving this to the la nina part.

**Author's response**: We will revise the manuscript as suggested.

**Reviewer comment:** L524: "tie" → "tied"

**Author's response**: We will revise the manuscript as suggested.

**Reviewer comment:** L550-551: "ENSO impacts West Antarctica through modulation of the ASL via Rossby wave propagation, though the ASL's influence on East Antarctica remains unclear", please add a reference to this.

**Author's response**: Reference will be added to this statement.

**Reviewer comment:** L583-585: Consider reformulating this sentence.

**Author's response**: We will revise the manuscript as suggested.

**Reviewer comment:** L595: The reference to Fig. 1c seems to be larger than the surrounding text.

**Author's response**: We will revise the manuscript as suggested.

**Reviewer comment:** L631: "However, the timescale of the response of the upstream ice to the positive SAM forcing is unclear and would involve a substantial lag". Please also describe how substantial this lag is what it would mean to the GRACE signal that you have used in this study.

**Author's response**: We will revise the manuscript to include a brief discussion of the potential lag in the response of upstream ice to positive SAM forcing. This lag, which may range from months to several years depending on regional ice dynamics, suggests that the GRACE signal may reflect a delayed response rather than an immediate reaction to SAM variability. We will clarify this point to help interpret the relationship between SAM and GRACE-derived mass changes more accurately.

**Reviewer comment:** L649: "This dynamical signal is stronger in West than in East Antarctica.". Add a citation to this.

**Author's response**: We will revise the manuscript as suggested.

**Reviewer comment:** L 658-659: The authors should add the time period that is considered in this study here. Also mention that you used ERA5 and RACMO.

**Author's response**: We will revise the manuscript as suggested.

**Reviewer comment:** L676-683: As it is the last concluding paragraph of the paper, remove references to figures and citations in this paragraph.

**Author's response**: We will revise the manuscript as suggested.

**Reviewer comment:** L676-683: Similar to my comment about the abstract, consider to shortly mention the uncertainties that have been discussed, such as the relatively short time period that you use and the various teleconnections that may have not happened yet within this time period, or other processes like atmospheric rivers.

**Author's response**: We will include it in the conclusion.

**Reviewer comment:** L690: This citation does not lead to the correct RACMO2.3p2 SMB data, as it refers to a newer version of RACMO: RACMO2.3p3.

**Author's response**: We will make the necessary correction.

---

## Author Comment (AC2)

RC 2.

**SUMMARY**

"The changing mass of the Antarctic Ice Sheet during ENSO-dominated periods in the GRACE era (2002-2022)" presents a comprehensive analysis of the circulation, surface mass balance, and ice mass variation patterns associated during four different periods of El Nino and La Nina phases of ENSO over two decades. The study ties together a number of prior studies on how ENSO impacts Antarctic surface mass balance by highlighting that the spatial impacts of this mode of variability vary strongly depending on the periods considered. It brings together observational, reanalysis, and model datasets to produce a compelling argument that the ENSO signal in Antarctica is dependent on event-specific atmospheric circulation patterns. I look forward to the publication of this manuscript; however, I have some major comments about the presentation of results without indications of statistical significance, the structure of the results, and the wording around association versus causation when establishing the occurrence of circulation and SMB/mass variability patterns during periods of El Nino and La Nina. Please see major and minor comments below.

**Author's response**: We appreciate your constructive feedback and believe that your suggestions will significantly enhance the clarity and scientific rigor of our study. We have carefully addressed each of the major and minor comments you raised.

**MAJOR COMMENTS**

**Reviewer comment:**

Statistical significance of trends and anomalies – many of the figures and corresponding analyses in this manuscript describe trends and anomalies in circulation, surface mass balance, and short-term mass change of the Antarctic Ice Sheet. However, the figures and discussion are missing critical information on the statistical significance of the results shown. For example, Fig. 2 shows the linear trend in ice mass change based on GRACE data, and here it would be very useful to add hatching or another indicator of where the trend is statistically significant. For Fig. 3, does the regression output p-values? If so, this would be another example of where it would be important to show where the statistically significant regions are. Same for Fig. 4 and 5 - for the composite maps, it would be key to add an indication for where the mean anomaly in sea level pressure is statistically significant (or exceeds the standard deviation among the different anomalies, for example). Without an indication on the maps for which regions exhibit statistically significant anomalies, readers cannot know which patterns are robust.

**Author's response**: We agree with the reviewer that indicating statistical significance is important for a robust interpretation of our results. We are currently implementing statistical significance tests for the trends and anomalies presented in the manuscript. Significant regions will be highlighted on the maps to help readers identify which observed patterns are robust.

**Reviewer comment**

For the analyses of figure 4 and 5, I recommend structuring the text either by region (then compare different periods) or by period (and go through each region). The current structure of the text alternates between period and region, and that makes it hard to follow.

**Author's response**: We will also restructure the text to enhance flow and readability, following your suggestion to present each region individually before making comparisons across regions.

**Reviewer comment**

There are several instances of language that implies causation rather than correlation throughout the paper. For example on L229, "the results show that ENSO influences circulation over Antarctica, driving short-term fluctuation in AIS mass…" – rather, the results show that ENSO periods are correlated with certain meridionally-oriented circulation patterns conducive to the flow of marine air masses onto the AIS. Furthermore, since there is not an analysis of the individual events that are contributing precipitation during the time periods in question, I would avoid using the word "driving" when it comes of the ENSO phase/circulation pattern and the associated SMB signals. As mentioned later in the text, precipitation can be driven by a few impactful events or many smaller snowfall events, or a mix of the two, and this study does not address the link between individual snowfall events and the large-scale circulation patterns. Furthermore, some of the language such as "that weakened the Antarctic high" or "a developing low-pressure system" or "leading to…" implies that this study examined the time-evolution of sea level pressure anomalies during the periods in question. My understanding of the methods is that this was not done – in which case, I would strongly recommend to the authors to remove any suggestions of the temporal evolution of anomalies throughout the text, unless there are figures to back up the claims.

**Author's response**: Regarding the language used, we will refine it to avoid implying causation or temporal evolution that is not supported by our methods. Although much of this language was revised in earlier drafts, we acknowledge that some instances still remain. We will carefully review the manuscript to ensure that all wording clearly reflects correlation rather than causation and avoids terms that may suggest otherwise.

**Reviewer comment**

L421-426 – I would be careful presenting the March 2022 event here as if it were the only extreme event/atmospheric river that occurred here over the time period studied. Certainly, this event was a standout and had a huge impact on the surface. At the same time, there are multiple atmospheric rivers impacting each location along the Antarctic coastline every year – meaning that there is the opportunity to assess the relationship between extremes, ENSO, and SAM. I would encourage the authors to discuss their results in the context of Shields et al. 2022 (https://agupubs.onlinelibrary.wiley.com/doi/full/10.1029/2022GL099577) – which examined the associated between different modes of variability and atmospheric river occurrence and precipitation. Please see Fig. 3 of the Shields paper in reference to L565-566 of the Discussion as well – which shows the correlation between atmospheric river days and negative SAM.

**Author's response**: We will ensure that our discussion remains measured and is clearly framed within the context of Shields et al. (2022) and related work.

**MINOR COMMENTS**

**Reviewer comment:** Abstract – would recommend removing/reducing the number of acronyms, including AIS, ASL, SAM, and SST.

**Author's response**: We will reduce the number of acronyms used throughout the manuscript for improved readability.

**Reviewer comment:** L17 – "… we investigate AIS mass variability" (add mass? Same for L26)

**Author's response**: Mass will be added to this line.

**Reviewer comment:** L22 – "anticyclonic circulation anomalies" (add circulation)

**Author's response**: We will add the term "circulation" as suggested.

**Reviewer comment:** L23-26 – sentence is a bit confusing, consider shortening or clarifying

**Author's response**: The sentence will be rewritten to improve clarity.

**Reviewer comment:** L27 – what does "event-scale" mean? Synoptic-scale?

**Author's response**: In this instance, "event-scale" refers to synoptic-scale events. We recognize that the term "event-scale" may cause confusion given our use of the term "ENSO events" elsewhere in the manuscript. To avoid ambiguity, we will revise the text and change it to "synoptic-scale".

**Reviewer comment:** L43 – Add "The" to beginning of sentence, and "is regionally dependent and affects different regions" is redundant

**Author's response**: We will edit the manuscript as suggested.

**Reviewer comment:** L57 – it may be helpful to mention Pacific South American mode 1 (PSA1) in the Introduction, since this is another term used to describe the second most-dominant mode of variability around Antarctica, associated with ENSO.

**Author's response**: We will edit the manuscript as suggested.

**Reviewer comment:** L65 – impact of ASL on East Antarctica – is there any evidence that the ASL influences East Antarctic circulation? This is also mentioned at the end of the manuscript, and I think it would be helpful to clarify (a) whether any links have been found between the ASL and East Antarctic circulation (to support the statement that "the impact" exists) and (b) what those links could be.

**Author's response**: There is no clear evidence of a direct ASL impact in East Antarctica; however, it is possible that the ASL indirectly influences the region. Our analysis of the 2020–2022 La Niña period suggests that the ASL may have contributed to moisture inflow into Dronning Maud Land. We will revise the text to clarify this point and include supporting references, as suggested.

**Reviewer comment:** L73 – "reducing precipitation and SMB in West Antarctica" – please be specific about which regions of West Antarctica

**Author's response**: Okay, we will revise the language to be more regionally specific to improve clarity and accuracy.

**Reviewer comment:** L84-105 – really nice summary here, framing the motivation for this study in the context of prior literature

**Author's response**: We thankful to the reviewer.

**Reviewer comment:** L112 – clarify what COST-G RL-01 V0003 50km is, and please add a discussion either here or in the Discussion section about the spatiotemporal resolution of GRACE observations.

How well do these observations capture spatial variability in accumulation? Is there a tendency to under/overestimate surface mass balance anomalies given the 300km resolution?

**Author's response**: We will expand the discussion of the GRACE dataset to include a description of its effective spatial resolution (~300 km) and the implications for detecting surface mass balance anomalies.

**Reviewer comment:** L128 – Is the linear trend sufficient for capturing ice mass variation over 2002-2022? Is the 7-month moving median specifically applied for the linear trend removal, or do all results shown include the 7-month-averaged signals? Are there regions where the trend is/isn't statistically significant, by grid point? Is the trend removed everywhere or only where it is significant?

**Author's response**: A 7-month moving median is used to smooth the GRACE data before computing the ice mass variation trend over the period 2002–2022. No significance test was conducted, but it will be included in our future analysis. The results for ENSO-dominated periods include the 7-month moving median signals, which are then detrended to focus on variability. The choice of a 7 month filter follows King et al. 2023 and is a subjective choice to dampen GRACE month-to-month noise only.

**Reviewer comment:** L132 – do you know if there is a lag between the initiation of an El Nino or La Nina event and the teleconnection that impacts Antarctic surface mass balance? Do you know the timescale of the teleconnection?

**Author's response**: King et al 2003 looked at it and we can't resolve it with our method as it is likely ~6 months.

**Reviewer comment:** Fig. 1 – "shows the cumulatively summed normalised raw indices after which it is renormalized" – I'm having a hard time understanding what the method is.

**Author's response**: We will reword the methods section and corresponding figure captions to improve clarity, and we will take special care to ensure consistent and precise use of terminology throughout the manuscript.

**Reviewer comment:** Fig. 1 – please clarify what metrics where used to determine the ENSO phases shaded in (d) and (e). Also, I would recommend moving the legend from (c) to (a) and because there is no text labeling the figure axes, I'd recommend adding titles to each figure.

**Author's response**: We will revise the manuscript according to the suggestion.

**Reviewer comment:** L211/212 – "relative strengthening" and "relative weakening"

**Author's response**: We will modify as suggested.

**Reviewer comment:** Fig. 3 – how was the regression of 10m wind anomalies performed? For u and v separately, or did you use the wind vectors? For detrending the variables, did you use a linear trend? I think it would be helpful to have more information on the methods used here.

**Author's response**: The regression was performed separately for the **u** and **v** components. Detrending was done by removing the linear trend. Additional information will be included.

**Reviewer comment:** L240 – It could be helpful to readers if you present some Antarctic Ice Sheet-integrated SMB values when discussing the precipitation anomalies during El Nino and La Nina.

**Author's response**: Okay, we will consider.

**Reviewer comment:** L242 – in Fig. 3, the W. Antarctic winds look more along-shore than onshore except over the Antarctic Peninsula – can you clarify? As a general comment, it is quite difficult to see the wind vectors along the Antarctic coast, meaning it's not always clear if/when a figure supports the conclusions in the text about wind directions at the coast.

**Author's response**: To address this comment, we will increase the size of the wind vectors in the figure to enhance visibility, particularly along the Antarctic coast. This adjustment will help clarify the wind patterns discussed in the text and more effectively support the conclusions presented.

**Reviewer comment:** L273 – for the different periods of El Nino events presented, it would perhaps be helpful as added context to know whether these events were central or eastern.

**Author's response**: We agree that it would be helpful to indicate whether the El Niño–dominated periods included Central or Eastern Pacific events. Rather than assigning events on a month-by-month basis, we will refer to established classifications in the literature to identify which periods include Central or Eastern Pacific El Niño events. This will provide useful context without implying monthly resolution that our data do not support.

**Reviewer comment:** L274 – "representing a weakened an/or shifted ASL" rather than an actual high-pressure system" – how do you know? Do you have a figure to show this?

**Author's response:** We make this observation in reference to the climatology, which indicates a high-pressure system over Antarctica. This pattern is evident when we compute the climatology over the study period (2002–2022), although it is not shown in the manuscript.

**Reviewer comment:** L276 – "influencing meridional circulation, thus driving distinct spatial patterns in SMB" – could add a mention of "marine intrusions"/marine air masses here to link these two processes (the meridional circulation and the SMB)

**Author's response:** The manuscript will be revised to establish a clearer connection between the two processes.

**Reviewer comment:** L278 – "West Antarctica as two regions" – I'm very confused about what region is actually meant by the Amundsen Sea sector. Are you including all of Marie Byrd Land and the Ross coast in the Amundsen Sea? Where does the Bellingshausen fall? I would recommend adding region names to one of your early maps, and being very specific in your description of regional patterns.

**Author's response:** A more detailed map with additional regional labels will be included to allow for clearer and more specific descriptions of the regional patterns discussed in the text.

**Reviewer comment:** L280 – "different signs but broadly uniform" – I am slightly confused by the wording in this sentence

**Author's response:** The sentence will be restructured for better clarity.

**Reviewer comment:** L286 – "influences" – please use language of association and not causation

**Author's response:** We will review the manuscript carefully to ensure the language is precise and does not imply causation where only correlation is observed.

**Reviewer comment:** L296 – "… over the continent that weakened the Antarctic high" – again, use "associated with a weakened Antarctic high" or similar

**Author's response:** We will review the manuscript carefully to ensure the language is precise and does not imply causation where only correlation is observed.

**Reviewer comment:** L298 – "observed positive anomalies" – from GRACE?

**Author's response:** We will clarify the **o**bserved positive anomalies.

**Reviewer comment:** L298 – "A low-pressure anomaly" – I see a low-pressure anomaly all along the coast, but not specifically between these two sites?

**Author's response:** We will clarify the position of the low-pressure anomaly.

**Reviewer comment:** L301-307 – do you have a hypothesis for why this pattern occurred? Other modes of variability and/or teleconnections?

**Author's response:** Again, we will clarify why this pattern occurred and the potential hypothesis. Our hypothesis is linked to that El Nino dominating period coinciding with a central Pacific El Nino event.

**Reviewer comment:** L308 – "two distinct mass variability responses" – I've seen this wording several times in the text and there are only two possible responses, right? Mass gain or loss? Please clarify.

**Author's response:** We agree with the reviewer that there are fundamentally two possible responses—mass gain or mass loss. We will revise the wording throughout the manuscript to clarify this and avoid ambiguous phrasing such as "two distinct mass variability responses."

**Reviewer comment:** L327 – "western Dronning Maud Land" – please be specific about the region, and label on a map

**Author's response:** We will revise as suggested.

**Reviewer comment:** L333 – "southerly wind flow" and "northerly winds" – these are wind anomalies, right? If so, please refer to them as anomalies throughout the text. Also, these wind vectors are very hard to see in the figure. Perhaps I am misunderstanding the text, but I find it a bit confusing regarding the generating of "northerly winds into western regions, supporting slight positive anomalies". I expect northerly winds to occur on the eastern flank of the low-pressure anomaly and I also see a convergence of northerly and southerly winds at the coast.

**Author's response:** The winds described here are wind anomalies and will be consistently referred to as anomalies throughout the text in a revised document. The wind vectors will be replotted and enlarged where possible to improve visibility for the reader.

We acknowledge the confusion and appreciate the reviewer's observation. The low-pressure anomaly is located further west of the Dronning Maud Land coast, and the northerly winds anomaly is on the eastern flank of the low-pressure anomaly. The convergence zone is more prominent toward Enderby Land, where a slight positive mass anomaly is observed. We will clarify this description in the text, and we expect the wind patterns to be more easily interpreted with the improved figure.

**Reviewer comment:** L339 – "central-eastern Dronning Maud Land"

**Author's response:** Agree, we will change it to central-eastern Dronning Maud Land.

**Reviewer comment:** L340 – "mid-latitude blocking pattern" – I would not necessarily call a high-pressure anomaly a mid-latitude block, without first looking at the mid-upper level geopotential height patterns and sea level pressure (not the anomaly).

**Author's response:** We will revise the sentence to state that the feature resembles a mid-latitude block, but we have not explicitly categorized it as such.

**Reviewer comment:** L344-347 – this sentence is long and a bit confusing, recommend breaking it into two

**Author's response:** The sentence will be break into two for more clarity.

**Reviewer comment:** L345 – 4c or 4b?

**Author's response:** 4b. We will fix this.

**Reviewer comment:** L348 – I don't know that I see mid-latitude westerlies in 4c? (also these are wind anomalies, right?) – maybe more like the polar jet?

**Author's response:** These winds are anomalies, and the westerly jet observed primarily influences the AIS and the Southern Ocean, rather than the mid-latitudes as noted in the comment. It is more likely associated with the polar front jet. We will revise to indicate this.

**Reviewer comment:** L351 – "pressure anomalies" – specify low or high

**Author's response:** A weak high-pressure anomaly over the continent. We will revise to…

**Reviewer comment:** L351 – "developing" implies time-evolution

**Author's response:** The language will be revised to: "accompanied by a low-pressure anomaly in the adjacent ocean."

**Reviewer comment:** Fig. 5 - I am slightly concerned that the striping in Fig. 5k, for example, which extends all the way from the interior to the coast (especially because the patterns exhibit spatial continuity). I would recommend to the authors that they mask out the interior region most affected by the striping.

**Author's response:** We will take this into consideration and assess how to define the boundary of what should or should not be masked, in a way that does not obscure meaningful signals and is feasible to implement. If a suitable method can be identified, we will apply the masking. Otherwise, we will revise the figure caption to guide the reader's attention toward the more robust signals along the coast.

**Reviewer comment:** L373 – "strengthening" – implies time-evolution

**Author's response:** The language will be revised to: "reflecting an intensification and/or shift of the Amundsen Sea Low (ASL) (Fig. 5a–d)."

**Reviewer comment:** L378 – these low-pressure anomalies all look pretty elongated to me?

**Author's response:** The statement will be deleted in the revised version.

**Reviewer comment:** L379 – "enhanced southerly wind anomalies" – in 5d, I see northeasterly onshore wind anomalies and positive SMB here in RACMO2?

**Author's response:** The focus was on the Amundsen Embayment, which led to the generalization of southerly wind anomalies. However, during our defined 2020–2022 La Niña-dominated period, the wind anomalies across the embayment are more northeasterly, potentially transporting moisture onshore—particularly along the western part of the embayment. This likely explains the positive SMB signal seen in the RACMO data. We will revise the manuscript to reflect this more accurate description.

**Reviewer comment:** L386 – "potentially can support convection and positive mass anomalies" – reference for this?

**Author's response:** The statement was a deduction intended to explain the observed positive mass anomaly. We will include a suitable reference to support this explanation in the revised manuscript.

**Reviewer comment:** L400 – again, here it would be very helpful to show what the regions of statistically significant positive/negative SMB are on the RACMO2 SMB maps.

**Author's response:** Statistical significance tests will be performed to define....

**Reviewer comment:** L409 – "resulting in uniform northerly winds and positive mass anomalies" – are you talking about the coast only? From the figure I see westerly and northwesterly winds, not purely northerly – though I would re-iterate that the wind vectors are so small in the maps that they are really hard to see. Finally, also mentioning once more that if these are wind anomalies they should always be referred to as such and not presented as if they were the actual wind field.

**Author's response:** As noted previously, the font size in the wind plots will be increased to improve the visibility of wind direction. The term "northerly wind anomalies" was used in reference to the coastal region; however, further inland the anomalies exhibit more westerly to northwesterly flow. We will revise the text to consistently refer to these as wind anomalies throughout.

**Reviewer comment:** L413 – "two distinct" – again, there are only two possible SMB responses, right?

**Author's response:** The current wording suggests a range of possibilities; however, there are only two possible outcomes. We will revise the text to more accurately reflect this conclusion.

**Reviewer comment:** L419 – "deepening" implies temporal evolution

**Author's response:** The language will be revised to: "due to the unusually deep low-pressure anomaly in the Pacific."

**Reviewer comment:** L419-421 – these two features (low-pressure anomaly in the Pacific and wind anomalies over Wilkes Land) seem far apart spatially – I'm missing the connection here with respect to the circulation?

**Author's response:** The low-pressure anomaly appears to direct northwesterly winds toward Dronning Maud Land, rather than specifically over Wilkes Land. However, the co-occurrence of La Niña and a positive SAM phase seems to alter the atmospheric circulation pattern. Instead of the expected zonally symmetric flow over the Antarctic Ice Sheet (AIS), the circulation becomes more asymmetric, resulting in northerly to northwesterly wind anomalies over Wilkes Land. We will revise the paragraph to improve clarity and accurately reflect these circulations.

**Reviewer comment:** Fig. 6 – again, there needs to be information on the statistical significance of the patterns in this figure, which will presumably support the authors' claims that different ENSO events are associated with different circulation and surface mass balance patterns.

**Author's response:** We agree with this suggestion and will perform a statistical significance test to support the conclusions presented in this study.

**Reviewer comment:** L430 – Amundsen Sea sector and Marie Byrd Land

**Author's response:** We will edit the manuscript accordingly.

**Reviewer comment:** L446-447 – language suggests causation

**Author's response:** The sentence will be rewritten to suggest a correlation rather than causation.

**Reviewer comment:** L453 – might help to remind readers what the bi-polar pattern is

**Author's response:** We will edit the manuscript accordingly.

**Reviewer comment:** L454 – what is meant by "underlying"? Most common, strongest, dominant?

**Author's response:** The "underlying" used here represent the dominant ENSO impact.

**Reviewer comment:** L470 – "coastal easterlies" – could you clarify this? I see coastal westerly wind anomalies in 4a, c, and d.

**Author's response:** There is weaken coastal easterlies (actual winds), which is shown by the observed coastal westerly wind anomalies.

**Reviewer comment:** L479 – western Ross Sea sector is not mentioned earlier in the text, nor is the Ross ice shelf shown in any figures. Could you clarify what is meant here?

**Author's response:** More geographical regions will be included and mention the manuscript going forward.

**Reviewer comment:** L490 – "the anomalous response can be attributed to altered Rossby wave propagation" – surely Rossby wave propagation influences almost all ENSO-associated circulation patterns around Antarctica?

**Author's response:** We will revise to clarify and highlight the differences in the propagation pathways of the Rossby waves during Central versus Eastern Pacific. Rossby wave propagation influences ENSO-associated circulation patterns around Antarctica. However, the source location of Rossby wave trains tends to be 20°–30° farther west during Central Pacific El Niño events compared to Eastern Pacific El Niño events. These differing propagation pathways result in a westward and latitudinal shift of the ASL during Central El Niño events relative to Eastern El Niño events.

**Reviewer comment:** L524 – "isolating ENSO signals" – I would be careful with stating that you are isolating ENSO signals here, because as was already mentioned, there are a number of different weather patterns and extremes that occurred during the periods over which the circulation and SMB patterns were composited.

**Author's response:**  We will reword to prevent any potential confusion.

**Reviewer comment:** L525 – "convergence zone that enhances precipitation" – reference for this? And can you be specific about exactly where you see the convergence occurring? Do you see this in the actual wind fields too, not only the anomalies?

**Author's response:** References will be added, and the identified convergence zones will be cross-checked against the actual wind fields.

**Reviewer comment:** L4545-548 – reference?

**Author's response:** A reference will be added to support this statement.

**Reviewer comment:** L550 – "ASL's influence on East Antarctica remains unclear" – as mentioned earlier, this implies that there is an influence but we don't know what it is – is that the conclusion from Li et al. 2022, as cited?

**Author's response:** The influence of the ASL is primarily centred over West Antarctica but can also indirectly affect atmospheric circulation over East Antarctica. Zhang et al. (2021) analysed 500 hPa geopotential height anomalies during La Niña periods and suggested that the ASL facilitates moisture advection into East Antarctica. Similarly, our analysis of the 2020–2022 La Niña period shows a comparable pattern, with the ASL advecting north-westerly winds into Dronning Maud Land. Reference changed to Zhang et al. 2021.

**Reviewer comment:** L559 – can use "significant" if you show statistical significance of mass changes in the figure

**Author's response**: That was an incorrect use of the term 'significant' in this context.

**Reviewer comment:** L574-579 – it's probably important to add there that it's equally likely that certain modes of variability and their associated circulation patterns may be conducive to atmospheric river landfall in certain regions.

**Author's response:** We are in agreement with this suggestion and will incorporate it into the revised manuscript.

**Reviewer comment:** L598 – "structure of the westerlies was altered" implies causation, and refers to the winds rather than the wind anomalies.

**Author's response:** We will rephrase the wording to indicate a correlation rather than imply causation.

**Reviewer comment:** Discussion – general comment: this is a very long section, and while it is interesting, I think it comes across as somewhat redundant following the results and before the conclusion. I would recommend shortening it where possible, to make the section more concise and less repetitive.

**Author's response:** Thank you for the helpful suggestion. We agree that the Discussion section is a bit long and, at times, overlaps with the Results. We'll work on tightening the section to make it more concise and focused, while still capturing the key interpretations.

---

## Author Response (AR2)

**Author's Response to Editor Comments**

We thank the editor for pointing out the following points for us to consider. We have addressed those points. The updated line numbers in the revised manuscript are indicated in parentheses.

**Editor Comment** - L138: "The data are provided …"

**Response:** To address this point, we have revised the sentence for clarity by splitting it into two: "We used the GRACE and GRACE Follow-On data. The data are provided by the GFZ German Research Centre for Geosciences (Landerer et al., 2020)." (page 6, L132–133).

**Editor Comment** - L141: What does "small-small mass changes" mean?

**Response:** The intention was to highlight the limitations of GRACE's spatial resolution. To clarify this point, we have revised the sentence to read: "This relatively coarse resolution limits GRACE's ability to resolve or capture relatively small mass changes, particularly those associated with localised SMB anomalies." (page 6, L140–142).

**Editor Comment** - L769: "Our results support…"

**Response:** We have revised it as suggested "Our results support" (page 30, L770).

**Editor Comment** - L771-772: "Previous studies have demonstrated …" which studies?

**Response:** We have addressed this by citing a relevant study and revising the sentence to: "A previous study demonstrated a reduction in mass during El Niño and an increase during La Niña across the Peninsula (Sasgen et al., 2010)." (page 32, L772–774).

**Editor Comment** - L777: "to be influenced"

**Response:** We have revised it as suggested "to be influenced" (page 30, L778).

**Responses to Reviewers:**

We would like to note that recent revisions have affected the line numbering in our manuscript. As a result, some references in our earlier responses to the reviewers may no longer align with the updated line numbers. To address this, we have attached revised responses below that accurately reflect these changes, and we kindly ask that this updated version be considered our formal response to the reviewer comments.

**Author's responses to reviewers:**

The manuscript has been revised in accordance with the reviewers' suggestions, for which we are sincerely grateful, as their feedback has significantly strengthened the study. In response to comments concerning clarity, paragraph structure, language to avoid implying causation, and the

inclusion of significance testing to support the robustness of the signals, substantial revisions have been made. These changes are reflected in the updated line numbering throughout the manuscript.

**Reviewer 1**

**SUMMARY**

The study presents the impact that different ENSO-induced atmospheric circulation changes have on Antarctic ice sheet mass changes and analyze teleconnections with the southern annular mode. The authors show that there is strong event-to-event spatial variability between ENSO events using GRACE observed mass changes, regional climate model output and ERA5. This work fits well within the scope of the journal and provides a contribution to the field. The manuscript is generally well written, but some paragraphs can be somewhat lengthy. The following comments should help with solving the remaining issues before publication, with e.g. L1 referring to line 1.

**Author's response**: We sincerely thank you for your thoughtful and constructive comments, which have greatly contributed to improving the clarity, structure, and scientific rigor of the manuscript. All major and minor points raised have been carefully considered and addressed.

**General comments:**

**Reviewer comment**

- Recently, a new version of the regional climate model RACMO2.4p1 was published for the Antarctic ice sheet (Van Dalum et al., 2025, https://doi.org/10.5194/egusphere-2024-3728), which includes new physics (in particular relevant here are changes in precipitation). Importantly, RACMO2.4p1 also has a higher horizontal resolution of 11 km compared to the 27 km resolution used in RACMO2.3p2. Using the SMB of RACMO2.4p1 would improve the comparisons done in this study and I suggest the authors to use this version instead of RACMO2.3p2. RACMO2.4p1 data can be found here: https://doi.org/10.5281/zenodo.14217231

**Author's response**: We were primarily interested in the broad patterns associated with ENSO. With the improved parameterisation in the updated regional model RACMO2.4p1, we revised our results using SMB outputs from this version. Although the differences compared to RACMO2.3p2 were minimal, we chose to retain the updated results. These are described in the data section (page 8, L237–245) and are also reflected throughout the manuscript, including in the abstract (page 1, L19).

**Reviewer comment**

- In the manuscript, basal melting is mentioned but SMB and mass changes are not studied on the ice shelves, hence relating the results to basal melting is difficult. Therefore, consider to include ice shelves in the comparison with RACMO SMB in e.g. Fig. 3b and elsewhere, and if possible also for GRACE, or explain why that cannot be done. Furthermore, it is also interesting to see how the SMB changes over the major ice shelves for each ENSO period.

**Author's response**:  The mention of basal melting in this study is solely in the context of existing literature and is not considered in our analysis. To avoid confusion, we have removed any reference to basal melting from the results and limited it to the discussion section (page 27, L674–677). Additionally, the description of ENSO conditions in the discussion has been revised to exclude any mention of basal melting: "During El Niño conditions, a weakened ASL and reduced coastal easterlies allow westerly wind anomalies to bring marine air masses onshore, which enhance snowfall and mass accumulation through orographic lifting (Paolo et al., 2018; Huguenin et al., 2024). In contrast, La Niña conditions strengthen the ASL and intensify coastal easterlies, limiting moisture transport and reducing precipitation (Huguenin et al., 2024; Hosking et al., 2013)" (page 28, L708–717). Any remaining references to basal melting are strictly within the context of cited literature (page 36, L959–960; page 37, L983–987).

**Reviewer comment**

- I think it is valuable for this study to mention whether an ENSO event is central or eastern and discuss if and how such events differ, as it may explain some of the patterns that are identified in this study and therefore increase understanding. The authors shortly discuss the potential importance in the manuscript, like on L486-495, but I think a more in-depth analysis will improve the manuscript. Other work, like Macha et al. (2024), may provide information about whether an ENSO event is central or eastern, or it can be determined by following methods described by Ren and Jin (2011).

**Author's response**: We used cumulatively summed ENSO indices to define ENSO periods in order to capture the net influence of mass change during these times, which could include transitions between Central and Eastern Pacific events. This approach does not distinguish between specific El Niño event types or seasonal phases. Therefore, our focus is not on linking mass patterns to particular El Niño types, but rather on the overall observed mass change.

When comparing the cumulatively summed Central, Eastern, and Niño3.4 indices, we found they are highly correlated and difficult to separate from one another in a cumulative sense, at least within the data span available. This limitation is addressed in the methodology (page 6, L217–221), with a comparison figure included in the supplement (page 38, L1028-1032). It is also discussed in the manuscript: "It is important to state that our defined ENSO periods do not distinguish between El Niño types or seasonal phases but instead capture the net mass change over the entire period, providing broader context for ice sheet mass balance." (page 29, L748–750).

**Reviewer comment**

- Not all locations that are discussed in the manuscript are shown on a map, like the Wedell Sea, Ross Sea, location of the ASL or the various ocean sectors. Including the locations mentioned in the manuscript will improve clarity, making it easier to follow.

**Author's response**: As suggested, we have revised the manuscript to include more regional delineation on the maps—specifically in Figure 2—to improve clarity and ease of interpretation. In this figure, we now define the following regions: Antarctic Peninsula (AP), Bellingshausen Sea (BS), Amundsen Sea (AS), Amundsen Sea Low (ASL), Pacific Sector (PS), Ross Sea (RS), Indian Ocean (IO), Atlantic Sector (AS), Wilkes Land (WL), Enderby Land (EL), Dronning Maud Land (DML), Coats Land (CL), and Weddell Sea (WS). Due to the density of content in some of the later figures, we were unable to include all these regional labels elsewhere. The updated figure is located on page 10, L292.

**Reviewer comment**

- Including maps where the SMB changes are shown in percentage of the total SMB for the considered periods will help to understand how big the impact of ENSO/SAM is on the various regions that are considered, as some changes may seem large in for example high precipitation areas, while they are only relatively small. An alternative could be to report the integrated SMB values in Gt yr-1 for the ENSO events for the whole domain and smaller regions and compare them to the reference period.

**Author's response**: Our primary focus was on absolute mass change for each period, reflecting our emphasis on ice sheet mass change during individual ENSO periods. However, we have also included supplementary maps illustrating the relative impact of SMB changes, expressed as a percentage of the climatological mean SMB for each ENSO-dominated period (page 39, L1044; page 40, L1051). To generate these maps, we calculated the mean SMB for each period, compared it to the long-term climatological mean at each grid point, and expressed the difference as a percentage. While the main text continues to emphasize absolute mass change, we have added a line referencing the relative mass change on page 13 (L365-366) and page 19 (L508-509).

**Specific comments:**

**Reviewer comment:** L18: As you also use regional climate model output in your study, it should be mentioned in the abstract as well.

**Author's response**:  We have included the model output in the text (L18-19).

**Reviewer comment:** L23-26: "… and its influence on the ASL and the Southern Ocean circulation can be equally (and in some cases more) important to AIS variability." Please specify with respect to what or rephrase this sentence.

**Author's response**:  We have rephrased this sentence for improved clarity. It now reads: "In both East and West Antarctica, this study shows that the spatial impact of any given ENSO-dominant period can trigger distinct circulation patterns which can variably influence surface mass balance and ice mass changes" (page 1, L27–31).

**Reviewer comment:** Abstract: I think it is also important to shortly mention the uncertainties in the abstract that you also mention in the text, such as the relatively short time period that you use and the various teleconnections that may have not happened yet within this time period, or other processes like atmospheric rivers.

**Author's response**:  We have now included a line in the abstract addressing uncertainties: "However, uncertainties remain, as the mass variability observed during ENSO-dominant periods may not be solely attributed to ENSO, due to teleconnections that may not have fully developed or may have been masked by other processes" (page 1, L31–33).

**Reviewer comment:** L29-30: "The drivers of inter-annual to decadal Antarctic Ice Sheet (AIS) mass variability are complex and not yet fully understood". Please add a reference to this.

**Author's response**:  As suggested, we have now added a reference to support this statement: "IMBIE Team, 2018" (page 1, L37).

**Reviewer comment:** L35: Not only precipitation but also riming can add to the SMB.

**Author's response**: "Riming" has been included in the revised text (page 2, L42).

**Reviewer comment:** L43: Can you specify here what typically the time scale is that the SAM changes from positive to negative, or vice versa and why the SAM happens?

**Author's response**: We have elaborated on the timescale of SAM changes and provided further explanation of the underlying mechanisms driving these variations. The revised text now reads: "The SAM signal is driven by a combination of internal atmospheric dynamics and external forcings, including stratospheric ozone depletion, increases in greenhouse gases, and tropical teleconnections (Fogt and Marshall, 2020a). It varies on timescales from weeks to decades, and its influence on Antarctic precipitation is regionally dependent (Marshall et al., 2017). During the positive phase of SAM, the westerlies around 60° S strengthen, and the overall impact on the AIS is a net decrease in SMB (Marshall et al., 2017; Medley and Thomas, 2019). Conversely, the net influence of the negative phase of SAM on the AIS is an increase in SMB (Medley and Thomas, 2019; Marshall et al., 2017)" (page 2, L50–58).

**Reviewer comment:** L50: Is the total reduction of precipitation in the East AIS typically comparable to the precipitation increase in West Antarctica and the western Antarctic Peninsula? In other words, looking at the AIS as a whole, does a positive SAM increase or decrease the SMB?

**Author's response**: We have revised the paragraph to more clearly reflect the net impact of SAM on AIS SMB. The updated sentence now reads: "During the positive phase of SAM, the westerlies around 60° S strengthen, and the overall impact on the AIS is a net decrease in SMB (Marshall et al., 2017; Medley and Thomas, 2019). Conversely, the net influence of the negative phase of SAM on the AIS is an increase in SMB" (page 2, L53–58). We believe this revision clarifies the polarity of SAM's influence on AIS SMB.

**Reviewer comment:** L67-75: Please add the location of the ASL, sectors like the Pacific sector, Indian sector etc. and other names in a map (for example in Fig. 2), which would help visualize the processes described the paper.

**Author's response**: We have included additional geographical labels in Figure 2 to improve clarity, including the mean locations of the ASL and the Pacific, Atlantic, and Indian sectors (page 10, L292). The figure now defines the following regions: Antarctic Peninsula (AP), Bellingshausen Sea (BS), Amundsen Sea (AS), Amundsen Sea Low (ASL), Pacific Sector (PS), Ross Sea (RS), Indian Ocean (IO), Atlantic Sector (AS), Wilkes Land (WL), Enderby Land (EL), Dronning Maud Land (DML), Coats Land (CL), and Weddell Sea (WS).

**Reviewer comment:** L76-83: Mention here why your study is different than the studies that you mention.

**Author's response**: To address this comment, we have added a sentence clarifying how this study is distinct from other studies. It now reads: "In contrast, our study investigates the spatial impacts of multiple individual ENSO periods (as defined in our study), enabling an assessment of how AIS mass variability differs between events and capturing the diverse responses across the ice sheet, rather than a mean response" (page 3, L100–102).

**Reviewer comment:** L87: As GRACE observes mass changes, the mass loss due to processes like runoff and sublimation are also included in the signal and should be mentioned here, even though they are relatively small compared to discharge.

**Author's response**: We have now added the contributions of runoff and sublimation to the GRACE observed mass changes. The revised text now reads: "Although mass loss from runoff and sublimation is included in the GRACE signal, these components are relatively minor compared to discharge. Over the interannual timescales, atmospheric variability dominates the observed mass changes (King et al. 2023)" (Page 3, L106-109).

**Reviewer comment:** L139: Please mention that the index is normalized in Fig. 1a.

**Author's response**: We have now stated the index is normalised in Fig. 1a page 7 (L222).

**Reviewer comment:** L149: Also mention that the climate indices are detrended in Fig. 1c.

**Author's response**: We have revised the caption for Figure 1c to clarify that the indices are detrended (page 7, L222).

**Reviewer comment:** L155-161: Consider moving this paragraph such that it is mentioned before the paragraph of L148-154.

**Author's response**: The paragraph has been moved to page 5 (L178–184) and deleted from page 6 (L192–198) for improved flow and clarity.

**Reviewer comment:** L162-164: "...where the positive phase of ENSO dominates the negative ENSO phase until a positive peak in the cumulative index is reached...". I think that I know what the authors mean but consider reformulating this to improve clarity. Also, do you apply a minimum length that an ENSO period has to last?

**Author's response**: We have rewritten the sentence to improve clarity and included a minimum duration criterion. It now reads: "In this study, we defined El Niño-dominated periods as an interval during which the positive phase of ENSO persists and outweighs the negative phase, culminating in a positive peak. Similarly, La Niña-dominated periods are defined as intervals during which the negative phase outweighs the positive phase, culminating in a negative peak. Only ENSO periods with a minimum duration of 12 months were considered in our analysis" (page 6, L199–205).

**Reviewer comment:** Fig. 1: Please add a description to the Y-axis of the figures. In Figure 1.d, consider adding ENSO and in Figure 1.e SAM in the top of the figure, which would help reading the figure more quickly.

**Author's response**: We have labelled the Y-axis as Std. Dev" and added "ENSO (Cumulative summed)" to Fig. 1d and "SAM (Cumulative summed)" to Fig. 1e (page 7, L222).

**Reviewer comment:** Section 2.3: It has not been mentioned in the paper before why you want to use a regional climate model and why it is necessary, which should be explained in e.g. the introduction before explaining what regional climate model you are going to use.

**Author's response**: We have added a brief rationale for using a regional climate model in the introduction. It now reads: "Since GRACE observes total mass change without distinguishing between the individual components of the mass balance, we use SMB output from a regional climate model RACMO2.4p1 to assess the contribution of SMB to the spatial patterns detected by GRACE" (page 4, L123–126).

**Reviewer comment:** L189: "...at its lateral and ocean boundaries..." → at its lateral boundaries and SST and sea ice extent at the sea surface boundary...

**Author's response**:  We have revised accordingly "lateral boundaries and SST and sea ice extent at the sea surface boundary" (page 8, L241).

**Reviewer comment:** Section 2.4: The authors should mention here why it is necessary to use ERA5 over RACMO output for the 10 m wind speeds and sea level pressure.

**Author's response**:  We have provided justification for using ERA5 pressure and wind variables instead of RACMO output. The revised text states: "We used ERA5 products instead of RACMO outputs because ERA5 provides broader spatial coverage and is more suitable for capturing large-scale atmospheric circulation patterns, which are critical for analysing ENSO-related teleconnections. Additionally, RACMO is forced by ERA5" (page 8, L254–257).

**Reviewer comment:** L225: Capital letter is missing in 'key'.

**Author's response**:  We have revised the caption accordingly, capitalizing "Key". The full caption now reads: "Figure 2. Linear rate and acceleration of AIS mass change (2002–2022) based on GRACE data from using univariate regression. Key Antarctic regions are labelled: Antarctic Peninsula (AP), Bellingshausen Sea (BS), Amundsen Sea (AS), Amundsen Sea Low (ASL), Pacific Sector (PS), Ross Sea (RS), Indian Ocean (IO), Atlantic Ocean (AO), Wilkes Land (WL), Enderby Land (EL), Dronning Maud Land (DML), Coats Land (CL), and Weddell Sea (WS). Stippling indicates areas not statistically significant ($p < 0.05$). Significance tests do not reflect the effects of temporal correlations in these data (Williams et al., 2014)" (page 10, L294-300).

**Reviewer comment:** L227-229: Also mention here that you plot ERA5 and RACMO in Figure 3.

**Author's response**:  We have included references to ERA5 and RACMO2.4p1 in both the main text and the Figure 3 caption. The text now states: "Figure 3 presents the regression results of cumulatively summed anomalies in ERA5 reanalysis climate variables (sea level pressure and 10 m winds) and RACMO2.4p1 SMB" (page 10, L302–303). The updated caption reads: "Figure 3. Maps show the regression of cumulatively summed sea level pressure (shaded region and contour) and 10 m wind anomalies (represented by reference vectors (m s⁻¹) from ERA5 reanalysis (a), cumulatively summed RACMO2.4p1 model SMB anomalies (b), and GRACE ice mass change anomalies (c)" (page 11, L314–320).

**Reviewer comment:** Fig. 3: I do not fully understand what is shown here. Is this the SLP and winds, SMB and GRACE mass loss averaged over the ENSO events? If this is the average over the ENSO events, including both El Nino and La Nina, would they not compensate each other?

**Author's response**: The figure shows the regression coefficients of sea level pressure (SLP) and wind anomalies (cumulatively summed), surface mass balance (SMB, cumulatively summed), and GRACE anomalies onto the cumulatively summed ENSO index. We have edited the Figure 3 caption to improve clarity: "Figure 3. Maps show the regression of cumulatively summed sea level pressure (shaded region and contour) and 10 m wind anomalies (represented by reference vectors (m s⁻¹)) from ERA5 reanalysis, cumulatively summed RACMO SMB anomalies, and GRACE ice mass change anomalies regressed against cumulatively summed Niño3.4. The u and v wind components were regressed separately. All panels reflect regressions of anomalies over the period 2002–2022. All

variables were linearly detrended prior to regression using the full data periods. Stippling indicates regions where the regression results are not statistically significant ($\rho<0.05$)" (page 11, L314–320).

**Reviewer comment:** Fig. 4 and 5: Interpreting the results would be easier if you mention in this figure for each ENSO event whether the SAM index is positive, negative or neutral.

**Author's response**: We have indicated the phase of the cumulatively summed SAM for each ENSO-dominated period in Figures 4 and 5 to aid interpretation. The updated Figure 4 now appears on page 15 (L368), and Figure 5 on page 21 (L515).

**Reviewer comment:** Fig. 4i-l: Do you know why the north-south striping is so much more pronounced in Fig. 4j and Fig. 4l compared to Fig. 4i and Fig. 4k?

**Author's response**: The north-south stripping is much more pronounced over shorter periods of time as there is less averaging. Furthermore, due to instrument degradation toward the end of the GRACE mission, the observational error increases, which likely contributes to the more noticeable north–south striping in Fig. 4j and Fig. 4l. Adding the significance hatching has helped with this interpretation Fig 4 is now on page 15 L368.

**Reviewer comment:** L310: Do you mean Fig. 4g instead of Fig. 4c?

**Author's response**: It is now Fig. 4g instead, with the update reflected on page 17 (L440). Additionally, the paragraph has been revised and differs from the earlier version. It now reads: "Positive SMB (Fig. 4e, g) and ice mass anomalies (Fig. 4j, l) are observed during the 2002–2005 and 2014–2016 El Niño periods, particularly in GRACE (Fig. 4i, k), whereas negative mass anomalies are evident during the 2009–2010 and 2018–2020 periods (Fig. 4j, l)."

**Reviewer comment:** L311-312: "Note that the 2002-2005 SMB anomaly is only marginally positive (Fig. 4a)." → Note that the 2002-2005 SMB anomaly is only marginally positive for the Antarctic Peninsula (Fig. 4e).

**Author's response**: We have revised the entire paragraph for clarity as suggested; therefore, the original sentence is no longer included (page 17, L435–436).

**Reviewer comment:** L313, 314: Fig. 4f → Fig 4f, h and also Fig. 4j → Fig. 4j, l.

**Author's response**:  We have revised as suggested (page 17, L442).

**Reviewer comment:** L323: Please also show these sectors on a map, e.g. Fig. 2.

**Author's response**:  We have included several geographical sectors on the map (Fig. 2) (page10, L292).

**Reviewer comment:** L330-353: Link the pressure anomalies and wind changes to moisture transport and their consequent impact on SMB and mass changes. These paragraphs can also be shortened.

**Author's response**:  In line with the reviewer's comment to avoid language implying causation rather than correlation—and given that our analysis does not directly show moisture transport—we have limited the discussion of links between pressure anomalies and moisture transport in the results section and moved that content to the discussion. Additionally, we have summarized the paragraph for clarity. The specific line referenced has been deleted (page 18, L465–476).

**Reviewer comment:** L380-381: Fig. 5f, g-h → Fig 5f-h and also Fig. 5j, k-l → Fig. 5j-l

**Author's response**: We have revised as suggested "Fig. 5f-h and Fig. 5j-l" (page 22, L552).

**Reviewer comment:** L385-387: Can you explain more how the northerly winds from the Pacific and southerly winds from the continent can lead to convection? And how it may result in positive mass anomalies?

**Author's response**: After revisiting our analysis, we observed that the pressure anomalies during the 2007–2009 La Niña period are not robust at the 0.05 significance level. Therefore, we cannot directly link these pressure anomalies to the observed mass gain, particularly in GRACE. We have revised the text to reflect this, which now reads: "In contrast, during the 2007–2009 La Niña period, a mass gain is prominently observed in GRACE (Fig. 5i), a pattern more commonly associated with El Niño periods described earlier. However, the SMB and pressure anomaly patterns during this period are not statistically significant at the 0.05 level" (page 23, L555–557).

**Reviewer comment:** L393-398: Similarly, as before, link the pressure and wind anomalies to moisture transport and then to SMB and mass changes.

**Author's response**: As mentioned earlier, in response to the reviewer's comment, we have limited the discussion of links between circulation, moisture transport, and mass changes in the results section and moved all related content to the discussion. This ensures we avoid drawing conclusions not directly supported by our analysis, as we did not examine moisture transport. The revised paragraph now reads: "This contrasting mass change response between the two periods aligns with the position of the negative pressure anomaly in the Pacific sector. In the 2010–2014 La Niña period, the pressure anomaly is centred over the Bellingshausen Sea, accompanied by offshore wind anomalies over the Peninsula (Fig. 5b). In contrast, during the 2020–2022 La Niña period, the negative pressure anomaly is centred in the Amundsen Sea, with onshore wind anomalies directed into the Peninsula (Fig. 5d)" (page 23, L574–579).

**Reviewer comment:** L421-426: How much of the 2020-2022 La Nina SMB signal is caused by this atmospheric river event? Is it possible that it is (almost) completely dominated by it?

**Author's response**: We explored the analysis by comparing the inclusion and exclusion of the March 2022 event and observed a signal in Wilkes Land, which has now been incorporated into the discussion. The revised text reads: "To determine the extent of the influence of this event, we examined the 2020–2022 period by comparing the inclusion and exclusion of the March 2022 event (Supplementary Fig. S5). While the March 2022 event increased the strength of the SMB positive anomaly in Wilkes Land, the region still observed a strong positive SMB anomaly during the 2020–2022 period when March 2022 was excluded (Supplementary Fig. S5). According to Wang et al. (2023), extreme events in March 2022 and October 2021 accounted for approximately 38% of the precipitation anomalies in Wilkes Land during the 2020–2022 La Niña period, driven by a pair of symmetrically distributed high–low pressure systems over the Southern Ocean near 120°W and 60°E" (page 32, L853–865). The corresponding supplementary figure is provided on page 41 (L1059).

**Reviewer comment:** Fig. 6: How did you calculate the average of the anomalies shown here? Did you weigh them by the length of the El Nino or La Nina-dominated periods? Or did you simply take the average of the maps that you have shown in Fig. 4 and 5?

**Author's response**: We simply took the average of the maps shown in Figures 4 and 5 and have clarified this in the main text (page 25, L633) and in the figure caption (page 26, L650). We are under the impression that the mean computed for the SLP/wind fields and the regression results for SMB and ice mass have already accounted for the length of each El Niño/La Niña period represented in Figures 4 and 5.

**Reviewer comment:** 459-461: Can you elaborate about these unusual climate dynamics? Does this have any impact on ENSO/SAM related SMB changes that you have discussed in the paper?

**Author's response**: We have added a line elaborating on the unusual climate dynamics and their influence on large-scale circulation (Xin et al., 2023), particularly the Southern Annular Mode (SAM), and how this may potentially affect ENSO teleconnections and their impact on AIS mass variability. This content has been relocated to page 36 (L942–944) and now reads: "However, between 2000 to 2020, shifts in large-scale circulation, particularly in SAM, have been reported, potentially affecting ENSO teleconnections and their influence on AIS variability."

**Reviewer comment:** L474-476: I am not sure if I fully understand how your results support the findings that increased basal melt is compensated by higher SMB. If I am not mistaken, you do not include ice shelves in your analysis where basal melt can occur, so how do you know that the positive SMB anomalies and increased mass that you show compensate for increased basal melt?

**Author's response**: To avoid confusion, we have revised the text and deleted the original line. The new paragraph now reads: "During El Niño conditions, a weakened ASL and reduced coastal easterlies allow westerly wind anomalies to bring marine air masses onshore, which enhance snowfall and mass accumulation through orographic lifting (Paolo et al., 2018; Huguenin et al., 2024). In contrast, La Niña conditions strengthen the ASL and intensify coastal easterlies, limiting moisture transport and reducing precipitation (Huguenin et al., 2024; Hosking et al., 2013)" (page 28, L708–717).

**Reviewer comment:** L477: "… El Nino-dominated period in the Amundsen sector differ" → "… El Nino-dominated periods in the Amundsen sector differs"

**Author's response**: To summarise and shorten the discussion, we have rewritten the paragraph and deleted the earlier version (page 28, L727–728). The revised paragraph now reads: "However, the 2009–2010 El Niño period deviates from this pattern, with negative SMB anomalies observed in the Amundsen Sea sector (Fig. 4f)" (page 28, L719–720).

**Reviewer comment:** L483-485: As you include the complete events, doesn't it make your methods more vulnerable for irregular events, such as atmospheric rivers, that may overshadow the ENSO signals?

**Author's response**: We have included in the discussion the limitations of our cumulative indexing method, specifically its vulnerability to high-frequency and short-term impacts. The revised text now reads: "However, across individual ENSO periods, the AIS response exhibits considerable variability, with each period associated with distinct atmospheric circulation patterns. It is possible that the teleconnection between tropical ENSO signals and Antarctic climate may not be fully established during a given ENSO phase or masked by other processes. Our analysis, which uses cumulative summed indices to match GRACE mass time series, is primarily sensitive to low-frequency variability and does not resolve shorter-term impacts, such as tropical convection pulses that initiate Rossby wave trains or high-frequency variability linked to storm systems like atmospheric rivers. Nonetheless, the integrated effect of these processes is captured by GRACE" (page 36-37, L971–978).

**Reviewer comment:** L508-510: Considering moving this to the la nina part.

**Author's response:** We have deleted this section and combined the discussion of La Niña and El Niño to improve clarity and cohesion (page 30, L768–770).

**Reviewer comment:** L524: "tie" → "tied"

**Author's response:** We have revised the text and combined the discussion of La Niña and El Niño for both West Antarctic and East Antarctic regions. As a result, the content has been updated across pages 28–30 (L694–782).

**Reviewer comment:** L550-551: "ENSO impacts West Antarctica through modulation of the ASL via Rossby wave propagation, though the ASL's influence on East Antarctica remains unclear", please add a reference to this.

**Author's response:** We have revised the text and deleted this section (page 34, L866–869). Additionally, much of the surrounding discussion has been updated to enhance clarity and reduce overall length.

**Reviewer comment:** L583-585: Consider reformulating this sentence.

**Author's response:** We have rewritten this section; therefore, the original content has been deleted (page 34, L897–902).

**Reviewer comment:** L595: The reference to Fig. 1c seems to be larger than the surrounding text.

**Author's response:** We have reduced the font size, now on page 32 (L854).

**Reviewer comment:** L631: "However, the timescale of the response of the upstream ice to the positive SAM forcing is unclear and would involve a substantial lag". Please also describe how substantial this lag is what it would mean to the GRACE signal that you have used in this study.

**Author's response:** We have briefly discussed the potential lag in the response of upstream ice to positive SAM forcing and how this is reflected in our results (page 36, L958–960).

**Reviewer comment:** L649: "This dynamical signal is stronger in West than in East Antarctica.". Add a citation to this.

**Author's response:** We have added the reference (Rignot et al., 2019) to support the statement (page 37 (L987).

**Reviewer comment:** L 658-659: The authors should add the time period that is considered in this study here. Also mention that you used ERA5 and RACMO.

**Author's response:** We have revised the text as suggested. The updated sentence now reads: "To examine the AIS mass change during different ENSO-dominated periods, we analysed AIS mass change anomalies observed by GRACE/GRACE-FO spanning the period 2002–2022. These anomalies were interpreted alongside RACMO2.4p1 modelled SMB and mean sea level pressure and 10 m winds from ERA5 reanalysis products" (page 37, L996–1000).

**Reviewer comment:** L676-683: As it is the last concluding paragraph of the paper, remove references to figures and citations in this paragraph.

**Author's response:** We have revised as suggested with Fig. 3 removed (Page 38, L1016) and Macha et al., 2024 also removed (page 38, L1025).

**Reviewer comment:** L676-683: Similar to my comment about the abstract, consider to shortly mention the uncertainties that have been discussed, such as the relatively short time period that you use and the various teleconnections that may have not happened yet within this time period, or other processes like atmospheric rivers.

**Author's response**: We have included a statement in the conclusion acknowledging the uncertainties in our analysis. The added text reads: "We acknowledge uncertainties in our analysis due to the relatively short ENSO-dominated periods considered. Some ENSO-related teleconnections may not have fully developed during these intervals, and other processes—such as atmospheric rivers—may have masked or modulated the ENSO signal, complicating the attribution of the observed spatial impacts" (page 38, L1018–1021).

**Reviewer comment:** L690: This citation does not lead to the correct RACMO2.3p2 SMB data, as it refers to a newer version of RACMO: RACMO2.4p1.

**Author's response**:  We have included a new link to RACMO2.4p1 to support accessibility and transparency (page 42, L1073).

RC 2.

**SUMMARY**

"The changing mass of the Antarctic Ice Sheet during ENSO-dominated periods in the GRACE era (2002-2022)" presents a comprehensive analysis of the circulation, surface mass balance, and ice mass variation patterns associated during four different periods of El Nino and La Nina phases of ENSO over two decades. The study ties together a number of prior studies on how ENSO impacts Antarctic surface mass balance by highlighting that the spatial impacts of this mode of variability vary strongly depending on the periods considered. It brings together observational, reanalysis, and model datasets to produce a compelling argument that the ENSO signal in Antarctica is dependent on event-specific atmospheric circulation patterns. I look forward to the publication of this manuscript; however, I have some major comments about the presentation of results without indications of statistical significance, the structure of the results, and the wording around association versus causation when establishing the occurrence of circulation and SMB/mass variability patterns during periods of El Nino and La Nina. Please see major and minor comments below.

**Author's response**:  We appreciate your constructive feedback and believe your suggestions have greatly contributed to improving the clarity and scientific rigor of our study. We have carefully addressed each of the major and minor comments you raised and incorporated the recommended changes throughout the manuscript.

**MAJOR COMMENTS**

**Reviewer comment:**

Statistical significance of trends and anomalies – many of the figures and corresponding analyses in this manuscript describe trends and anomalies in circulation, surface mass balance, and short-term mass change of the Antarctic Ice Sheet. However, the figures and discussion are missing critical information on the statistical significance of the results shown. For example, Fig. 2 shows the linear trend in ice mass change based on GRACE data, and here it would be very useful to add hatching or another indicator of where the trend is statistically significant. For Fig. 3, does the regression output p-values? If so, this would be another example of where it would be important to show where the

statistically significant regions are. Same for Fig. 4 and 5 - for the composite maps, it would be key to add an indication for where the mean anomaly in sea level pressure is statistically significant (or exceeds the standard deviation among the different anomalies, for example). Without an indication on the maps for which regions exhibit statistically significant anomalies, readers cannot know which patterns are robust.

**Author's response**: We have implemented statistical significance tests for the trends and anomalies presented in this study. Significant regions are shown without stippling, while non-significant regions are stippled to help readers identify where the observed patterns are robust. A description of the significance testing approach has been added to the Methods section (page 6, L217–221). We further note here that the interpretation of this needs to be done in the context of our use of cumulative values of each of mass, winds and pressure and so the meaning of significance is different to using mass rates, or unmodified winds and pressure. The significance test is performed for the pressure, SMB and GRACE anomalies, no significance test was performed for the wind vectors.

**Reviewer comment**

For the analyses of figure 4 and 5, I recommend structuring the text either by region (then compare different periods) or by period (and go through each region). The current structure of the text alternates between period and region, and that makes it hard to follow.

**Author's response**: Our current structure for presenting the results in Figures 4 and 5 is organised by ENSO period. We present El Niño and La Niña periods separately, and within each period, we discuss the regions in sequence: starting with West Antarctica, followed by the Antarctic Peninsula, and then East Antarctica. However, we are unsure how this differs from the recommendation to structure the results "by period and go through each region."

**Reviewer comment**

There are several instances of language that implies causation rather than correlation throughout the paper. For example, on L229, "the results show that ENSO influences circulation over Antarctica, driving short-term fluctuation in AIS mass..." – rather, the results show that ENSO periods are correlated with certain meridionally-oriented circulation patterns conducive to the flow of marine air masses onto the AIS. Furthermore, since there is not an analysis of the individual events that are contributing precipitation during the time periods in question, I would avoid using the word "driving" when it comes of the ENSO phase/circulation pattern and the associated SMB signals. As mentioned later in the text, precipitation can be driven by a few impactful events or many smaller snowfall events, or a mix of the two, and this study does not address the link between individual snowfall events and the large-scale circulation patterns. Furthermore, some of the language such as "that weakened the Antarctic high" or "a developing low-pressure system" or "leading to..." implies that this study examined the time-evolution of sea level pressure anomalies during the periods in question. My understanding of the methods is that this was not done – in which case, I would strongly recommend to the authors to remove any suggestions of the temporal evolution of anomalies throughout the text, unless there are figures to back up the claims.

**Author's response**: Regarding the language used, we have refined the manuscript to avoid implying causation or temporal evolution that is not supported by our methods. We carefully reviewed the text to ensure that all wording clearly reflects correlation rather than causation and avoids

terminology that may suggest otherwise. As a result, substantial revisions have been made in the latest edition compared to the earlier version.

**Reviewer comment**

L421-426 – I would be careful presenting the March 2022 event here as if it were the only extreme event/atmospheric river that occurred here over the time period studied. Certainly, this event was a standout and had a huge impact on the surface. At the same time, there are multiple atmospheric rivers impacting each location along the Antarctic coastline every year – meaning that there is the opportunity to assess the relationship between extremes, ENSO, and SAM. I would encourage the authors to discuss their results in the context of Shields et al. 2022 (https://agupubs.onlinelibrary.wiley.com/doi/full/10.1029/2022GL099577) – which examined the associated between different modes of variability and atmospheric river occurrence and precipitation. Please see Fig. 3 of the Shields paper in reference to L565-566 of the Discussion as well – which shows the correlation between atmospheric river days and negative SAM.

**Author's response**:   We explored the impact of the March 2022 event by comparing analyses with and without its inclusion. This revealed a notable signal in Wilkes Land, which we have incorporated into the discussion. Specifically, we added the following: "To determine the extent of the influence of this event, we examined the 2020–2022 period by comparing the inclusion and exclusion of the March 2022 event (Supplementary Fig. S5). While the March 2022 event increased the strength of the SMB positive anomaly in Wilkes Land, the region still observed a strong positive SMB anomaly during the 2020–2022 period when March 2022 was excluded (Supplementary Fig. S5). According to Wang et al. (2023), extreme events in March 2022 and October 2021 accounted for approximately 38% of the precipitation anomalies in Wilkes Land during the 2020–2022 La Niña period, driven by a pair of symmetrically distributed high–low pressure systems over the Southern Ocean near 120°W and 60°E." (page 32-33, L853–865; Supplementary Fig. S5 on page 41, L1058-1062).

**MINOR COMMENTS**

**Reviewer comment:** Abstract – would recommend removing/reducing the number of acronyms, including AIS, ASL, SAM, and SST.

**Author's response**:   We have reduced the number of acronyms used in the abstract to improve readability. Specifically, we removed ASL, SAM, and SST, and retained acronyms only for terms that appear more than once. These changes are reflected on page 1 (L13–34).

**Reviewer comment:** L17 – "… we investigate AIS mass variability" (add mass? Same for L26)

**Author's response**:   We have included "mass" to now become "We investigated AIS mass variability" (page 1, L17).

**Reviewer comment:** L22 – "anticyclonic circulation anomalies" (add circulation)

**Author's response**:   We have added the term "circulation" as suggested (page 1, L26).

**Reviewer comment:** L23-26 – sentence is a bit confusing, consider shortening or clarifying

**Author's response**:   We have rephrased the sentence for improved clarity. It now reads: "In both East and West Antarctica, this study shows that the spatial impact of any given ENSO-dominant period can trigger distinct circulation patterns which can variably influence surface mass balance and ice mass changes" (page 1, L27–31).

**Reviewer comment:** L27 – what does "event-scale" mean? Synoptic-scale?

**Author's response**:  To avoid ambiguity, we have deleted the entire sentence from the text (page 1, L33–34).

**Reviewer comment:** L43 – Add "The" to beginning of sentence, and "is regionally dependent and affects different regions" is redundant

**Author's response**:  We have added "The" at the beginning of the sentence on page 2 (L50) and removed the redundant part on page 2 (L53).

**Reviewer comment:** L57 – it may be helpful to mention Pacific South American mode 1 (PSA1) in the Introduction, since this is another term used to describe the second most-dominant mode of variability around Antarctica, associated with ENSO.

**Author's response**:  We have included a brief introduction of PSA-1 in the ENSO section to clarify its role in transmitting ENSO signals to Antarctica. The added text reads: "This Rossby wave train leads to the formation of the Pacific South American mode 1 (PSA-1), an atmospheric anomaly pattern that enables ENSO signals to reach Antarctica (Hoskins and Karoly, 1981). This creates a positive-pressure anomaly over the Amundsen–Bellingshausen sector (ABS) during El Niño events—the positive phase of PSA-1—and a negative-pressure anomaly during La Niña conditions—the negative phase of PSA-1 (Turner, 2004; Hoskins and Karoly, 1981)" (page 2, L69–73).

**Reviewer comment:** L65 – impact of ASL on East Antarctica – is there any evidence that the ASL influences East Antarctic circulation? This is also mentioned at the end of the manuscript, and I think it would be helpful to clarify (a) whether any links have been found between the ASL and East Antarctic circulation (to support the statement that "the impact" exists) and (b) what those links could be.

**Author's response**:  No direct link has been found between ASL and East Antarctic circulation, and this remains an area of active research. To avoid overstating the connection, we have deleted the relevant sentence from the introduction (page 3, L80) and the discussion (page 33, L877–880).

**Reviewer comment:** L73 – "reducing precipitation and SMB in West Antarctica" – please be specific about which regions of West Antarctica

**Author's response**: We have added specific locations to the sentence for improved clarity. It now reads: "over the Antarctic Peninsula and from the Bellingshausen Sea to the Ross Sea region in West Antarctica" (page 3, L88–89).

**Reviewer comment:** L84-105 – really nice summary here, framing the motivation for this study in the context of prior literature

**Author's response**: We thank the reviewer.

**Reviewer comment:** L112 – clarify what COST-G RL-01 V0003 50km is, and please add a discussion either here or in the Discussion section about the spatiotemporal resolution of GRACE observations. How well do these observations capture spatial variability in accumulation? Is there a tendency to under/overestimate surface mass balance anomalies given the 300km resolution?

**Author's response**: We have expanded the description of the GRACE dataset to clarify what COST-G RL-01 V0003 represents. Specifically, we now state: "We used the COST-G release 1 version 3 (RL-01 V0003) gridded mass anomaly product, which combines GRACE/GRACE-FO solutions from multiple GRACE analysis centres" (page 4, L136–138). Additionally, we address the limitations of GRACE's original spatial resolution (~300 km), noting: "This relatively coarse resolution limits GRACE's ability

to resolve or capture relatively small mass changes, particularly localised surface mass balance anomalies" (page 4, L140–141).

**Reviewer comment:** L128 – Is the linear trend sufficient for capturing ice mass variation over 2002-2022? Is the 7-month moving median specifically applied for the linear trend removal, or do all results shown include the 7-month-averaged signals? Are there regions where the trend is/isn't statistically significant, by grid point? Is the trend removed everywhere or only where it is significant?

**Author's response**: All results shown for GRACE and RACMO were smoothed using a 7-month moving median window. This choice, following King et al. (2023), was a subjective decision aimed at reducing month-to-month noise in the GRACE data. The linear trend was then computed from the smoothed data.

To address your question regarding the sufficiency of a linear trend, we have now included an acceleration term and tested it for statistical significance (page 10, L292). In response to the reviewer comment, we have also added indications of statistical significance to the relevant figures.

Figures 3, 4, 5, and 6 are computed from detrended data to focus on variability.

**Reviewer comment:** L132 – do you know if there is a lag between the initiation of an El Nino or La Nina event and the teleconnection that impacts Antarctic surface mass balance? Do you know the timescale of the teleconnection?

**Author's response**: King et al 2023 looked at it and found a ~6-month lag - we can't resolve it with our method.

**Reviewer comment:** Fig. 1 – "shows the cumulatively summed normalised raw indices after which it is renormalized" – I'm having a hard time understanding what the method is.

**Author's response**: We have reworded the figure caption to improve clarity to: "Figure 1. Monthly climate indices of SAM (Marshall, 2003) and Niño3.4 from 2002–2022: (a) normalised SAM and Niño3.4 indices; (b) normalised cumulatively summed SAM and Niño3.4 indices; (c) detrended, cumulatively summed SAM and Niño3.4 indices (normalised). Periods until positive peaks are reached in the cumulatively summed Niño3.4 are defined as El Niño-dominated and La Niña-dominated periods, respectively, represented as red and blue shaded areas in (d). Similarly, periods until positive and negative peaks are reached in the cumulatively summed SAM index (Marshall, 2003) are defined as SAM-positive and SAM-negative dominated periods, respectively, denoted as red and blue shaded areas in (e). Neutral dominated periods are represented by white shading." (page 7, L224–236).

**Reviewer comment:** Fig. 1 – please clarify what metrics where used to determine the ENSO phases shaded in (d) and (e). Also, I would recommend moving the legend from (c) to (a) and because there is no text labeling the figure axes, I'd recommend adding titles to each figure.

**Author's response**: Titles and metrics have been added to Figure 1 to improve clarity. However, due to limited white space in panel (a), we have retained the legend in panel (c) (page 7, L222). To determine the ENSO phases shaded in panel (d), we applied the threshold: "Only ENSO periods with a minimum duration of 12 months were considered in our analysis," as described in the Methods section (page 6, L204).

**Reviewer comment:** L211/212 – "relative strengthening" and "relative weakening"

**Author's response**: Relative has been added as suggested (page 8, L267-268).

Reviewer comment: Fig. 3 – how was the regression of 10m wind anomalies performed? For u and v separately, or did you use the wind vectors? For detrending the variables, did you use a linear trend? I think it would be helpful to have more information on the methods used here

**Author's response**: Additional information has been included in the caption to clarify the regression method for the 10 m wind anomalies and the detrending approach. The updated caption reads: "Figure 3. Maps show the regression of cumulatively summed sea level pressure (shaded region and contour) and 10 m wind anomalies (represented by reference vectors [m s⁻¹] from ERA5 reanalysis), cumulatively summed RACMO SMB anomalies, and GRACE ice mass change anomalies regressed against cumulatively summed Niño3.4. The u and v wind components were regressed separately. All panels reflect regressions of anomalies over the period 2002–2022. All variables were linearly detrended prior to regression using the full data period. Stippling indicates regions where the regression results are not statistically significant ($p < 0.05$)." (page 11, L314–320).

**Reviewer comment:** L240 – It could be helpful to readers if you present some Antarctic Ice Sheet-integrated SMB values when discussing the precipitation anomalies during El Nino and La Nina.

**Author's response**: We have included mean SMB anomalies for El Niño and La Niña years across various regions of the Antarctic Ice Sheet. These values are based on composites of annual accumulation anomalies during ENSO years and are now presented in the text (page 12, L333–339).

**Reviewer comment:** L242 – in Fig. 3, the W. Antarctic winds look more along-shore than onshore except over the Antarctic Peninsula – can you clarify? As a general comment, it is quite difficult to see the wind vectors along the Antarctic coast, meaning it's not always clear if/when a figure supports the conclusions in the text about wind directions at the coast.

**Author's response**: We have increased the size of the wind vectors in Figure 3 to enhance visibility, particularly along the Antarctic coast. This adjustment improves the clarity of the wind patterns discussed in the text. As wind vectors are not tested for statistical significance, we have also removed portions of the results discussion that relied on detailed interpretation of coastal wind direction (page 12, L322–332).

**Reviewer comment:** L273 – for the different periods of El Nino events presented, it would perhaps be helpful as added context to know whether these events were central or eastern.

**Author's response**: This point was also raised by Reviewer 1. As noted in our earlier response, the cumulative method we use may encompass multiple El Niño event types, making it difficult to distinguish between central and eastern Pacific events. We defined ENSO periods using cumulatively summed ENSO indices to capture the net influence on mass change over time, which may include transitions between event types and seasonal phases. Our approach does not separate individual El Niño types or seasonal timing.

Therefore, our focus is on the observed mass change rather than attributing it to specific El Niño classifications. We compared the cumulatively summed Central, Eastern, and Niño3.4 indices and found them to be highly correlated, making it difficult to separate their effects in a cumulative framework, especially given the length of the data record. This limitation is now addressed in the methodology (page 6, L217–221), with a comparison figure included in the Supplement (page 38, L1029), and discussed in the main text: "It is important to state that our defined ENSO periods do not distinguish between El Niño types or seasonal phases but instead capture the net mass change over the entire period, providing broader context for ice sheet mass balance." (page 29, L748–750).

**Reviewer comment:** L274 – "representing a weakened an/or shifted ASL" rather than an actual high-pressure system" – how do you know? Do you have a figure to show this?

**Author's response:** We based this interpretation on the climatology over the study period (2002–2022), although no figure is shown. To avoid confusion, we have revised the relevant section to read: "In West Antarctica, El Niño-dominated periods are characterised by a positive pressure anomaly in the Pacific sector off the West Antarctic coastline (Fig. 4a–b). The position and strength of these positive pressure anomalies vary for each El Niño-dominated period, which is also reflected in the variation of wind anomalies and spatial patterns of SMB (Fig. 4e–h) and ice mass change (Fig. 4i–l). However, during the 2018–2020 period, no significant pressure anomaly is observed, and in the 2009–2010 period, a significant pressure anomaly is located closer to the continent, with a non-significant pressure anomaly further north (Fig. 4a–b)." (page 16, L383–390).

**Reviewer comment:** L276 – "influencing meridional circulation, thus driving distinct spatial patterns in SMB" – could add a mention of "marine intrusions"/marine air masses here to link these two processes (the meridional circulation and the SMB)

**Author's response:** We have added to the discussion which is where it is more appropriate in regard to your previous comment on causation "allow westerly wind anomalies to bring marine air masses, onshore, which, enhance snowfall and mass accumulation through orographic lifting" on page 28 L710-711.

**Reviewer comment:** L278 – "West Antarctica as two regions" – I'm very confused about what region is actually meant by the Amundsen Sea sector. Are you including all of Marie Byrd Land and the Ross coast in the Amundsen Sea? Where does the Bellingshausen fall? I would recommend adding region names to one of your early maps and being very specific in your description of regional patterns.

**Author's response:** To avoid confusion, we have deleted the sentence referring to "West Antarctica as two regions" (page 16, L391–392). Additionally, we have updated Figure 2 to include detailed regional labels, allowing for clearer and more specific descriptions of the spatial patterns discussed in the text (page 10, L292).

**Reviewer comment:** L280 – "different signs but broadly uniform" – I am slightly confused by the wording in this sentence

**Author's response:** To address the confusion, we have deleted the entire paragraph containing the phrase "different signs but broadly uniform" (page 16, L393).

**Reviewer comment:** L286 – "influences" – please use language of association and not causation

**Author's response:** We have revised the language throughout this section to remove any implication of causation and instead emphasize correlation. We have deleted the sentence containing "influences" (page 16, L387).

**Reviewer comment:** L296 – "… over the continent that weakened the Antarctic high" – again, use "associated with a weakened Antarctic high" or similar

**Author's response:** To avoid implying causation, we have revised the paragraph to reflect association rather than direct influence. The updated text now reads: "For the 2014–2016 El Niño-dominated period, we observed weak and, in some regions, non-significant positive SMB and ice mass anomalies in the Amundsen Sea sector and western Ross Sea (Fig. 4g, k). During this period, our cumulative ENSO and SAM were out of phase (El Niño/+SAM), as evidenced by significant negative

pressure anomalies over the continent (Fig. 4c). The positive pressure anomaly in the Pacific was located away from the coastline and was associated more with wind anomalies along the shore, rather than onshore." (page 17, L410–420).

**Reviewer comment:** L298 – "observed positive anomalies" – from GRACE?

**Author's response:** Clarified to observed positive "SMB and ice mass" anomalies and rewritten as "For the 2014–2016 El Niño-dominated period, we observed weak and, in some regions, non-significant positive SMB and ice mass anomalies in the Amundsen Sea sector and western Ross Sea (Fig, 4g, k)." Page 17 L410-411.

**Reviewer comment:** L298 – "A low-pressure anomaly" – I see a low-pressure anomaly all along the coast, but not specifically between these two sites?

**Author's response:** We have reworded the sentence to more accurately describe the spatial extent of the pressure anomaly. It now reads: "During this period, our cumulative ENSO and SAM were out of phase (El Niño/+SAM), as evidenced by significant negative pressure anomalies over the continent (Fig. 4c)." (page 17, L414–415).

**Reviewer comment:** L301-307 – do you have a hypothesis for why this pattern occurred? Other modes of variability and/or teleconnections?

**Author's response:** Our potential hypothesis is that the mass change observed during this period may be primarily driven by the Central Pacific El Niño event. We now discuss this in the revised discussion section, referencing Macha et al. (2024):
"However, the 2009–2010 El Niño period deviates from this pattern, with negative SMB anomalies observed in the Amundsen Sea sector (Fig. 4f). The pressure anomaly during this period is distinct, with a positive pressure anomaly extending from the Amundsen Sea to beyond the Ross Sea. An important difference from other El Niño periods is the westward extension of this positive pressure anomaly, which reduces moisture transport into the region. This period encompasses a strong Central Pacific El Niño event (Kim et al., 2011), and the associated pressure anomaly (Fig. 4b) resembles patterns linked to such events, which are associated with moisture-depleted wind anomalies and suppressed precipitation in the Amundsen and Bellingshausen regions (Chen et al., 2023; Macha et al., 2024)." (page 28, L719–726)
"Our 2009–2010 El Niño mass pattern aligns with Macha et al. (2024), who reported reduced accumulation during Central Pacific El Niño events in the SON and JJA seasons. These similarities suggest that the observed mass change may reflect the impact of Central Pacific El Niño phases during the SON and JJA seasons in the Amundsen Sea sector. It is important to state that our defined ENSO periods do not distinguish between El Niño types or seasonal phases but instead capture the net mass change over the entire period, providing broader context for ice sheet mass balance." (page 29, L744–747)

**Reviewer comment:** L308 – "two distinct mass variability responses" – I've seen this wording several times in the text and there are only two possible responses, right? Mass gain or loss? Please clarify.

**Author's response:** We have reworded the phrase to avoid confusion, replacing "two distinct mass variability responses" with "contrasting mass change" (page 17, L432).

**Reviewer comment:** L327 – "western Dronning Maud Land" – please be specific about the region, and label on a map

**Author's response:** We have addressed this comment by providing a detailed map in Figure 2, which includes regional labels and boundaries to support clearer spatial references. Additionally, we have

reworded the relevant section to offer a more specific description of the region in question (page 18, L454–464).

**Reviewer comment:** L333 – "southerly wind flow" and "northerly winds" – these are wind anomalies, right? If so, please refer to them as anomalies throughout the text. Also, these wind vectors are very hard to see in the figure. Perhaps I am misunderstanding the text, but I find it a bit confusing regarding the generating of "northerly winds into western regions, supporting slight positive anomalies". I expect northerly winds to occur on the eastern flank of the low-pressure anomaly, and I also see a convergence of northerly and southerly winds at the coast.

**Author's response:** Yes, that is correct. We have modified the text to clarify that these are wind anomalies and have emphasized this terminology throughout the manuscript. Additionally, the rest of the text has been edited to consistently refer to wind anomalies, ensuring clearer interpretation of the figures and associated patterns.

**Reviewer comment:** L339 – "central-eastern Dronning Maud Land"

**Author's response:** We have chosen to focus on the most robust signals in this region. As a result, much of this section has been rewritten to emphasize significant and large-scale patterns, rather than more localized or ambiguous signals such as those in central-eastern Dronning Maud Land.

**Reviewer comment:** L340 – "mid-latitude blocking pattern" – I would not necessarily call a high-pressure anomaly a mid-latitude block, without first looking at the mid-upper level geopotential height patterns and sea level pressure (not the anomaly).

**Author's response:** To avoid confusion, we have deleted the term "mid-latitude blocking pattern" and reworded the entire sentence. It now reads: "Conversely, during the 2009–2010 El Niño period, we observed a significant anomalous mass gain in Dronning Maud Land (Fig. 4f, j). This mass gain coincides with a significant positive pressure anomaly over the Atlantic, which supports onshore wind anomalies into Dronning Maud Land." (page 18, L476–479).

**Reviewer comment:** L344-347 – this sentence is long and a bit confusing, recommend breaking it into two

**Author's response:** The sentence has been rewritten for clarity and readability. It now reads: "In the Indian Ocean sector/Wilkes Land, mass gain is broadly observed during the 2002–2005 and 2009–2010 El Niño periods (Fig. 4e, f, i, j), and a reduction in mass during the 2014–2016 and 2018–2020 El Niño periods (Fig. 4g, h, k, l). During the periods with mass gain, positive pressure anomalies were present over Wilkes Land (Fig. 4a, b), with the anomaly more intense and statistically significant during the 2009–2010 El Niño period and associated with a greater magnitude of mass gain in Wilkes Land (Fig. 4b, f, j). Conversely, during periods broadly associated with mass reduction (Fig. 4g, h, k, l), negative pressure anomalies were observed around the Wilkes Land region, aligned with offshore wind anomalies across much of the sector (Fig. 4c, d)." (page 19, L487–499).

**Reviewer comment:** L345 – 4c or 4b?

**Author's response:** Thank you for catching that. The correct reference is Figure 4b, and we have updated the citation accordingly (page 19, L495).

**Reviewer comment:** L348 – I don't know that I see mid-latitude westerlies in 4c? (also these are wind anomalies, right?) – maybe more like the polar jet?

**Author's response:** To address the confusion and improve clarity, we have deleted the entire paragraph in question (page 19, L500–505).

**Reviewer comment:** L351 – "pressure anomalies" – specify low or high

**Author's response:** we have deleted the entire paragraph in question (page 19, L500–505).

**Reviewer comment:** L351 – "developing" implies time-evolution

**Author's response:** We have revised the wording to avoid implying time-evolution. The updated sentence now reads: "Conversely, during periods broadly associated with mass reduction (Fig. 4g, h, k, l), negative pressure anomalies were observed around the Wilkes Land region, aligned with offshore wind anomalies across much of the sector (Fig. 4c, d)." (page 19, L497–499).

**Reviewer comment:** Fig. 5 - I am slightly concerned that the striping in Fig. 5k, for example, which extends all the way from the interior to the coast (especially because the patterns exhibit spatial continuity). I would recommend to the authors that they mask out the interior region most affected by the striping.

**Author's response:** We are cautious about removing regions without a consistent and repeatable method. Instead, we now test for statistical significance, and the striping has been heavily masked out using stippling. Also, we guide the reader in the figure caption to focus on the more robust signals along the coast (page 21–22, L515–528).

**Reviewer comment:** L373 – "strengthening" – implies time-evolution

**Author's response:** We have revised the language to avoid implying time-evolution. The updated sentence now reads: "Overall, during our La Niña-dominated periods, the Pacific sector exhibits a persistent negative pressure anomaly (Fig. 5a–d), which appears more elongated than the positive pressure anomaly associated with El Niño periods." (page 22, L531-532).

**Reviewer comment:** L378 – these low-pressure anomalies all look pretty elongated to me?

**Author's response:** To address the concern, we have deleted the original statement and revised the text to reflect a more balanced interpretation: "Overall, during our La Niña-dominated periods, the Pacific sector exhibits a persistent negative pressure anomaly (Fig. 5a–d), which appears more elongated than the positive pressure anomaly associated with El Niño periods." (page 22, L531-532).

**Reviewer comment:** L379 – "enhanced southerly wind anomalies" – in 5d, I see northeasterly onshore wind anomalies and positive SMB here in RACMO2?

**Author's response:** While there are some indications of positive SMB in RACMO2, the broader pattern reflects a reduction in mass. To clarify this, we have revised the entire paragraph to emphasize the dominant signals: "Three out of the four La Niña periods (2010–2014, 2016–2018, and 2020–2022) are broadly associated with negative SMB (Fig. 5f–h) and ice mass anomalies (Fig. 5j–l) across the Amundsen Sea sector. The reduction in mass during the 2020–2022 and 2010–2014 La Niña periods aligns with a significant negative pressure anomaly in the Pacific sector, and offshore wind anomalies (Fig. 5b, d)." (page 22, L551–554).

**Reviewer comment:** L386 – "potentially can support convection and positive mass anomalies" – reference for this?

**Author's response**: Based on our significance testing, the pressure anomalies during this period were not robust. Therefore, we have revised the sentence to avoid unsupported implications and references. It now reads: "In contrast, during the 2007–2009 La Niña period, a mass gain is prominently observed in GRACE (Fig. 5i), a pattern more commonly associated with El Niño periods described earlier. However, the SMB and pressure anomaly patterns during this period are not statistically significant at the 0.05 level." (page 23, L555–557).

**Reviewer comment:** L400 – again, here it would be very helpful to show what the regions of statistically significant positive/negative SMB are on the RACMO2 SMB maps.

**Author's response:** We have incorporated statistical significance tests into Figures 2, 3, 4, 5, and 6 to highlight regions of significant positive and negative SMB. The accompanying text has also been updated to reflect and guide interpretation of this additional information.

**Reviewer comment:** L409 – "resulting in uniform northerly winds and positive mass anomalies" – are you talking about the coast only? From the figure I see westerly and northwesterly winds, not purely northerly – though I would re-iterate that the wind vectors are so small in the maps that they are really hard to see. Finally, also mentioning once more that if these are wind anomalies they should always be referred to as such and not presented as if they were the actual wind field.

**Author's response:** We have reworded the paragraph to clarify the spatial patterns and avoid mischaracterising wind direction or implying actual wind fields. The revised text emphasizes wind anomalies and regional variability:

"Along the Atlantic sector, a dipole-like mass anomaly pattern is present during the 2007–2009 and 2020–2022 La Niña periods (Fig. 5e, h), whereas a more uniform response is observed during the 2010–2014 and 2016–2018 La Niña periods (Fig. 5f, g). During the 2007–2009 La Niña period, positive SMB anomalies were observed over Coats Land and negative SMB anomalies toward Enderby Land (Fig. 5e), with this spatial pattern reversed during the 2020–2022 La Niña period (Fig. 5h). Positive mass anomalies were also observed across the Atlantic region during the 2014–2016 La Niña period, with a reversed pattern during the 2016–2018 La Niña period. Regionally, Dronning Maud Land shows consistent positive SMB (Fig. 5f, h) and ice mass anomalies (Fig. 5j, l) during the 2010–2014 and 2020–2022 La Niña period." (page 23, L588–597)

**Reviewer comment:** L413 – "two distinct" – again, there are only two possible SMB responses, right?

**Author's response:** We have revised the wording to avoid implying multiple distinct types of SMB responses. The updated sentence now reads: "In the Indian Ocean sector/Wilkes Land, we found no consistent mass response to La Niña-dominated periods." (page 24, L607).

**Reviewer comment:** L419 – "deepening" implies temporal evolution

**Author's response:** To avoid implying temporal evolution, we have deleted this portion of the text (page 24, L623–630).

**Reviewer comment:** L419-421 – these two features (low-pressure anomaly in the Pacific and wind anomalies over Wilkes Land) seem far apart spatially – I'm missing the connection here with respect to the circulation?

**Author's response:** We have deleted this portion of the text (page 24, L623–630).

**Reviewer comment:** Fig. 6 – again, there needs to be information on the statistical significance of the patterns in this figure, which will presumably support the authors' claims that different ENSO events are associated with different circulation and surface mass balance patterns.

**Author's response:** Statistical significance test has been performed and added to the figure (page 26, L650).

**Reviewer comment:** L430 – Amundsen Sea sector and Marie Byrd Land

**Author's response:** Edited as suggested (page, 25 L637).

**Reviewer comment:** L446-447 – language suggests causation

**Author's response:** We have revised the language to avoid implying causation and as suggested, have restructured much of the discussion to provide a more concise summary. These changes are reflected on page 27, L660–666.

**Reviewer comment:** L453 – might help to remind readers what the bi-polar pattern is

**Author's response:** We have revised the text to briefly reiterate the meaning of the dipole (bi-polar) pattern (page 27, L663).

**Reviewer comment:** L454 – what is meant by "underlying"? Most common, strongest, dominant?

**Author's response:** This portion of the text has been removed from the discussion. The dipole pattern is now framed as a longer-timescale response, as reflected in the revised discussion on page 27, L662–663.

**Reviewer comment:** L470 – "coastal easterlies" – could you clarify this? I see coastal westerly wind anomalies in 4a, c, and d.

**Author's response:** We have clarified the description to accurately reflect the wind anomaly patterns observed. The revised sentence now reads: "During El Niño conditions, a weakened ASL and reduced coastal easterlies allow westerly wind anomalies to bring marine air masses onshore, which enhance snowfall and mass accumulation through orographic lifting." (page 28, L708–711).

**Reviewer comment:** L479 – western Ross Sea sector is not mentioned earlier in the text, nor is the Ross ice shelf shown in any figures. Could you clarify what is meant here?

**Author's response:** We have clarified the geographical reference by including detailed regional labels in Figure 2 and updating the text accordingly to reflect and define these regions, including the western Ross Sea sector and Ross Ice Shelf (page 10, L292).

**Reviewer comment:** L490 – "the anomalous response can be attributed to altered Rossby wave propagation" – surely Rossby wave propagation influences almost all ENSO-associated circulation patterns around Antarctica?

**Author's response:** We have revised text to remove any confusion "However, the 2009–2010 El Niño period deviates from this pattern, with negative SMB anomalies observed in the Amundsen Sea sector (Fig. 4f). The pressure anomaly during this period is distinct, with a positive pressure anomaly extending from the Amundsen Sea to beyond the Ross Sea. An important difference to the other El Niño periods is the extension of this positive pressure anomaly further to the west, which decreases moisture transport into the region. This period encompasses a strong Central Pacific El Niño event (Kim et al., 2011), and the associated pressure anomaly (Fig. 4b) resembles patterns linked to such

events, which are associated with moisture-depleted wind anomalies and suppressed precipitation in the Amundsen and Bellingshausen regions (Chen et al., 2023; Macha et al., 2024)." (page 28, L719–726).

**Reviewer comment:** L524 – "isolating ENSO signals" – I would be careful with stating that you are isolating ENSO signals here, because as was already mentioned, there are a number of different weather patterns and extremes that occurred during the periods over which the circulation and SMB patterns were composited.

**Author's response:** We have deleted it to prevent any potential confusion: "Similarly, the 2007–2009 La Niña period shows a mass pattern that contrasts with other La Niña periods, featuring a positive mass anomaly in the Amundsen Sea sector (Fig. 5i). However, atmospheric circulation patterns during this period do not statistically support the observed mass gain, suggesting that it may be linked to unrelated weather events or other modes of climate variability." (page 29, L751–754).

**Reviewer comment:** L525 – "convergence zone that enhances precipitation" – reference for this? And can you be specific about exactly where you see the convergence occurring? Do you see this in the actual wind fields too, not only the anomalies?

**Author's response:** We have deleted it prevent any potential confusion "Similarly, the 2007–2009 La Niña period shows a mass pattern that contrasts with other La Niña periods, featuring a positive mass anomaly in the Amundsen Sea sector (Fig. 5i). However, atmospheric circulation patterns during this period do not statistically support the observed mass gain, suggesting that it may be linked to unrelated weather events or other modes of climate variability." page 29 L751-754.

**Reviewer comment:** L4545-548 – reference?

**Author's response:** Thank you for pointing this out. We have added appropriate references to support the statement: "(Fogt et al., 2012; Fogt and Marshall, 2020b; Marshall et al., 2013)" (page 32, L830).

**Reviewer comment:** L550 – "ASL's influence on East Antarctica remains unclear" – as mentioned earlier, this implies that there is an influence, but we don't know what it is – is that the conclusion from Li et al. 2022, as cited?

**Author's response:** As there is no direct link established in Li et al. (2022), but only suggestions of a potential connection, we have deleted the statement to avoid implying a confirmed influence.

**Reviewer comment:** L559 – can use "significant" if you show statistical significance of mass changes in the figure

**Author's response**: Changed to anomalous (page 32, L833).

**Reviewer comment:** L574-579 – it's probably important to add there that it's equally likely that certain modes of variability and their associated circulation patterns may be conducive to atmospheric river landfall in certain regions.

**Author's response:** We have incorporated this point by noting that certain modes of variability and their associated circulation patterns may indeed be conducive to atmospheric river landfall in specific regions (page 32, L837–844). We have also expanded the discussion of uncertainties in our analysis to reflect this complexity (page 36-37, L971–978).

**Reviewer comment:** L598 – "structure of the westerlies was altered" implies causation and refers to the winds rather than the wind anomalies.

**Author's response:** We have reworded the text to avoid implying causation and now frame the discussion in terms of findings from other studies. These revisions are reflected on page 32, L853–865.

**Reviewer comment:** Discussion – general comment: this is a very long section, and while it is interesting, I think it comes across as somewhat redundant following the results and before the conclusion. I would recommend shortening it where possible, to make the section more concise and less repetitive.

**Author's response:** Thank you for the helpful suggestion. To improve clarity and reduce redundancy, we have substantially revised the discussion section to make it more concise and focused, while retaining the key insights and interpretations.

---

## Author Response (AR3)

We sincerely thank the editor and reviewers for their constructive feedback, which has been invaluable in shaping the manuscript to publication standard. We have carefully addressed all minor comments, as outlined below and incorporated into the revised manuscript, with changes reflected in the updated line numbering.

**Minor comments:**

L235 and elsewhere in the text, figures and captions: A climatology is usually defined as a 30-year period or longer. Here you examine the period 2002-2022 which is shorter than a 'climatology'. Please, write out the period extent rather than using the term climatology.

**Author's response:** In the revised manuscript, we have replaced the use of climatology with explicit reference to the study period (2002–2022) throughout the text, figures, and captions to avoid confusion.

L145 -146: This effectively produces mass anomalies with respect to the 2002–2022 GRACE period.

L238: depicted in the figures represent anomalies from the 2002–2022 period for each relevant.

L243: whereas negative anomalies indicate a reduction in mass relative to 2002–2022.

L248-249: statistical significance was assessed relative to the 2002–2022 baseline using a two-sample *t*-test assuming unequal variances.

Also, in the supplementary file.

L27-28: Figure S3. Map of RACMO2.4p1 SMB changes, expressed as a percentage relative to 2002–2022 during El Niño-dominated periods.

L35-36: Figure S4. Map of RACMO2.4p1 SMB changes, expressed as a percentage relative to 2002–2022 during La Niña-dominated periods.

**Reviewer comment:** L240: "reduction in mass relative to 2002-2022."

**Author's response:** We have revised the text as suggested, which now reads: "L243: reduction in mass relative to 2002–2022."

**Reviewer comment:** L250: Here you state that Fig. 2b shows a (quadratic) acceleration, but units are in "kg m-2 yr-1". This should rather be "kg m-2 yr-2". Please modify the units in Figure 2b if relevant.

**Author's response:** Figure 2b has been revised, and the acceleration is now expressed with the unit "$kg\ m^{-2}\ yr^{-2}$" (L265-266).

**Reviewer comment:** L281-282, 321, 394-395: Statistical significance is generally obtained for pval<0.05. Here you state the opposite, please clarify.

**Author's response:** Thanks for pointing it out. We have clarified it in the text as "(p > 0.05)" to show

that non-significant regions are stippled. This correction has been applied at L271, L286, L325, L399, L465 and L44 in the supplementary file.

**Reviewer comment:** L380: I guess the authors mean "(Supplementary Fig. 4)" here.

**Author's response:** Correct, we meant "(Supplementary Fig. 4)". However, based on your comment regarding maintaining consistency between "(Supplementary Fig. 4)" and "Fig. S4", we have edited the text as follows: "Absolute mass changes are shown in this section, while relative mass changes can be found in Fig. S4" L383-384.

**Reviewer comment:** L623: I am not sure to understand what the authors mean. Is the 22-year time series too short to fully sample the influence of ENSO-SAM on SMB variability? Please reformulate.

**Author's response:** The intention here is to state that the time span might not be sufficient to capture the full range of ENSO variability. The text has been revised to reflect this clearer.

"Given that our analysis spans a 22-year period, it is insufficient to capture the full range of ENSO variability, which requires a longer time period to be fully represented (Stevenson et al., 2010). Future studies should therefore consider a longer record , together with climate models, to better isolate and capture purely ENSO-driven signals" L630-633.

**Supplementary Figures:**

**Reviewer comment:** Across the text, refer to "Supplementary Fig. X" or "Fig. SX" and remain consistent.

**Author's response:** In line with your comment, to remain consistent we have revised the text with "Fig. SX". The revision can be found on L197, 288, 290 ,316, 384, 572, 574.

**Reviewer comment**: Supplementary Figure 1: Add labels a) and b) for individual subplots and refer to Supplementary Fig. 1a or b accordingly in the text and captions. Explicitly mention what "NEP" and "NCP" mean in the figure caption.

**Author's response:**  We have added labels a) and b) to the figure (Fig. S1) and accordingly refer to it in the text "Fig. S1b" L196.  The "NEP" and "NCP" meaning has been mentioned in now in the figure caption, which now reads "Figure S1. Timeseries of ENSO metric indices. The Niño Eastern Pacific (NEP, blue), Niño Central Pacific (NCP, red), and Niño3.4 (black) indices are shown" L7-8 in the supplementary file.

**Reviewer comment:** Supplementary Figure 2: Add labels a-d) and modify the figure caption accordingly.

**Author's response:** The figure has been updated and labels (a–d) have been added. The figure caption has been revised accordingly and now reads: "Figure S2. Composite maps showing the impact of El Niño and La Niña events on ERA5 mean sea level pressure (shading and contour, hPa) and 10 m wind anomalies (vectors, m s$^{-1}$), together with surface mass balance (SMB) anomalies (kg m$^{-2}$ y$^{-1}$) from RACMO2.4p1 over 2002–2022: (a, b) ERA5 mean sea level pressure and 10 m wind

anomalies for El Niño and La Niña, respectively; (c, d) SMB anomalies for El Niño and La Niña, respectively." This revision appears on L16–21 in the supplementary file.

**Reviewer comment:** Supplementary Figure 3: Add labels a-d) and modify the figure caption accordingly. In supplementary text 3, please list and provide the periods extent. I suggest refraining from using the term "climatological" for periods shorter than 30 years.

**Author's response:** The figure has been updated and labels (a–d) have been added. The figure caption has also been revised to reflect this and now reads: "Figure S3. Map of RACMO2.4p1 SMB changes, expressed as a percentage relative to 2002–2022 during El Niño-dominated periods: (a) Apr 2002–Aug 2005, (b) Mar 2014–Jun 2016, (c) Apr 2009–Apr 2010, and (d) Jun 2018–Apr 2020, corresponding to the defined El Niño-dominated periods." This revision appears on L27–29 in the supplementary file.

**Reviewer comment:** Supplementary Figure 5: Please provide a caption for this figure, including references to subplots using individual labels.

**Author's response:** A caption has been provided for Fig. S5, with references to the subplot labels in the supplementary file (L41–44): "Figure S5. Map showing the rate of change of SMB anomalies during the 2020–2022 La Niña-dominated period: (a) with the inclusion of the March 2022 extreme event, (b) with the March 2022 event excluded, and (c) the difference between (a) and (b)."

**Reviewer comment:** Supplementary Figure 4: Add labels a-d) and modify the figure caption accordingly. In supplementary text 4, please list and provide the periods extent. I suggest refraining from using the term "climatological" for a period shorter than 30 years.

**Author's response:** We have updated the figure and added labels (a–d). The figure caption has been revised to avoid the use of the term climatological and now reads: "Figure S4. Map of RACMO2.4p1 SMB changes, expressed as a percentage relative to 2002–2022 during La Niña-dominated periods: (a) Feb 2007–Apr 2009, (b) Jun 2016–Jun 2018, (c) Mar 2010–Mar 2014, and (d) Apr 2020–Dec 2022, corresponding to the defined La Niña-dominated periods." This revision appears on L34–37 in the supplementary file.

**Main Figures:**

**Reviewer comment:** Figure 2: Please add labels a) and b) for individual subplots and explicitly mention these labels in the figure caption as e.g. , "a) Linear rate and b) acceleration of AIS mass change …"

**Author's response:** The figure and caption have been revised to include labels. The figure caption now reads: "Figure 2. a) Linear rate and b) acceleration of AIS mass change (2002–2022) based on GRACE data using univariate regression." This revision appears on L267.

**Reviewer comment**: Figure 3 caption: "Regression of cumulatively summed…" Remove "(" between "anomalies" and "represented".

**Author's response:** We have revised as suggested and now reads "Figure 3. Regression of cumulatively summed sea level pressure (shaded region and contour) and 10 m wind anomalies represented by reference vectors (m s⁻¹) from ERA5 reanalysis (a), cumulatively summed RACMO2.4p1 model SMB anomalies (b), and GRACE ice mass change anomalies (c) regressed against

cumulatively summed Niño3.4. The u and v wind components were regressed separately. All panels reflect regression anomalies over the period 2002-2022. All variables were linearly detrended prior to regression using the full data periods. Stippling indicates regions where the regression results are not statistically significant (ρ>0.05). (L279-285)

**Reviewer comment**: Figure 4 caption: Replace "(left), (middle) or (right)" by proper labels "(a-d), (e-h) and (i-l)".

**Author's response**: Revised accordingly and the "(left), (middle) or (right)" has now been replaced with the proper labels "(a-d), (e-h) and (i-l)".

"Figure 4. Atmospheric circulation anomalies relative to the GRACE period (2002–2022) (a–d), rate of change in cumulative SMB anomalies from RACMO2.4p1 model (e–h) and linear rate of GRACE-derived ice mass anomalies (i–l) during El Niño-dominated period." (L317-320)

**Reviewer comment**: Figure 5 caption: Same comments as for Fig. 4.

**Author's response**: We have revised according to the previous comments. "Figure 5. Atmospheric circulation anomalies relative to the GRACE period (2002–2022) (a–d), rate of change in cumulative SMB anomalies from the RACMO2.4p1 model (e–h), and linear rate of GRACE-derived ice mass anomalies (i–l) during La Niña-dominated period." (L390-392)

**Reviewer comment**: Figure 6: Add labels a-f) and modify caption accordingly.

**Author's response:** Revised as suggested and labels have been added to the figure and caption. "Figure 6. The composites are generated based on the results of the four defined ENSO-dominated periods combined. ERA5 mean sea level pressure and 10 m wind anomalies (a–b), RACMO2.4p1 SMB (c–d), and GRACE-derived ice mass change (e–f)." (L459-462)

**References:**
**Reviewer comment:** L209: Add a reference to the RACMO2.4p1 data at the end of this sentence.

**Author's response:** A reference has been added to the end of this sentence. "We used modelled SMB output from the Regional Atmospheric Climate Model RACMO2.4p1 model (Van Dalum et al., 2025; Van Dalum et al., 2024)." (L211-212)

**Style:**
**Reviewer comment:** L53: "decades, meaning that the regional …"

**Author's response:** We have included "that" in the revised text "decades, meaning that the regional" L53.

**Reviewer comment:** L88: "…, rather than a mean signal."
**Author's response:** We have replaced "response" with signal **"rather than a mean signal"** L88.

**Reviewer comment:** L117: "… GRACE Follow On data, provided by the GFZ …"

**Author's response:** We have revised the sentence as follows: "We used the GRACE and GRACE Follow On data, provided by the GFZ German Research Centre for Geosciences (Landerer et al., 2020)" L117-118.

**Reviewer comment:** L123: "The data are provided on a 50 km grid …"

**Author's response:** We have corrected the spelling to "are," and the sentence now reads: "The data are provided on a 50 km grid." This revision appears on L123.

**Reviewer comment:** L133-134: "… redistribution are modelled in a similar fashion to spherical harmonic … 1 terms, i.e., based on the approach …"

**Author's response:** The text wasn't the correct description and has been revised to reflect the right description. "The effects of atmospheric and oceanic mass redistribution are modelled using standard de-aliasing products. Spherical harmonic degree-1 terms are added based on the approach of Swenson et al. (2008)." L133-135

**Reviewer comment:** L152: "… ENSO based on SST anomalies in the central …"

**Author's response:** Sea surface temperature has been abbreviated as SST, and the text now reads "ENSO based on SST anomalies in the central" L153.

**Reviewer comment:** L171: "… (Fig. 1b) and further detrended them (Fig. 1c)."

**Author's response:** We have inserted "further" and "them" in the sentence "(Fig 1b) and further detrended them (Fig. 1c)" L172.

**Reviewer comment:** L229-230: Remove the first sentence and start with "We use the term 'El Niño- or … or simply 'period' when considering periods of sustained ENSO phase as defined using our …"

**Author's response:** We have removed the first sentence and revised as follows: "We use the term 'El Niño- or La Niña-dominated period' or simply 'period' when considering periods of sustained ENSO phase as defined using our cumulatively summed index" L232-235.

**Reviewer comment:** L260: "large-scale"

**Author's response:** The error "large-scle" has been corrected to "large-scale." L263.

**Reviewer comment:** L290: "SMB and ice mass increase"

**Author's response:** The "s" at the end of "increases" has been deleted. "SMB and ice mass increase" L294.

**Reviewer comment:** L292-294: "Since SMB fluctuations are closely linked … ice mass change vary (Fig. 3b-c)."

**Author's response:** The sentence has been revised "Since SMB fluctuations are closely linked to ice mass change, the spatially coherent patterns between SMB and GRACE-derived ice mass change vary (Fig. 3b–c)" L296-298.

**Reviewer comment:** L295: "…, which indicates …"
**Author's response:** We have added "s" to the end of indicate "which indicates" L299.

**Reviewer comment:** L312: "(Supplementary Fig. 3)"

**Author's response:** For consistency we have replaced it with "Fig. S3" L316.

**Reviewer comment:** L336: "(Fig. 4g, k)"

**Author's response:** The comma has now been replaced with a period "(Fig. 4g, k)" L340.

**Reviewer comment:** L354: "show consistent patterns with negative SMB…"

**Author's response:** "Show" and "patterns" has been added to the sentence " show consistent patterns with negative SMB" L358.

**Reviewer comment:** L390: "illustrate variability in AIS mass …"

**Author's response:** "ASI" has been corrected to "AIS" L394.

**Reviewer comment:** L394: "pronounced north-south…"

**Author's response:** We have added "d" to the end of pronounce "pronounced north-south striping" L398.

**Reviewer comment:** L456: "mean sea level pressure"

**Author's response:** "Seal" has been corrected to "sea" and now reads "mean sea level pressure" L461.

**Reviewer comment:** L457: "mass change (e-f) "

**Author's response:** Labels have replaced "right" and now reads "mass change (e–f)" L461-462.

**Reviewer comment:** L459: "wind anomalies as vectors … SMB and GRACE data … Non-significant areas are stippled …"

**Author's response:** Revised as suggested " wind anomalies as vectors (m s$^{-1}$). SMB and GRACE data (kg m$^{-2}$ y$^{-1}$) are shown. Non-significant areas are stippled" L464-465.

**Reviewer comment:** L466: "pattern"

**Author's response:** We have inserted "r" for the correct spelling "pattern" L471

**Reviewer comment:** L467: "Antarctica during El Niño periods, and vice-versa during La Niña …"

**Author's response:** Revised accordingly "Antarctic during El Niño periods, and vice-versa during La Niña" L472.

**Reviewer comment:** L471: "SMB variability drives ice mass changes…"

**Author's response:** Revised as suggested and "s" has been added to "drive" and "of" deleted "SMB variability drives ice mass changes" L476.

**Reviewer comment:** L477: "(Figs. 4, 5)"

**Author's response:** Space inserted between Figs. And 4  "(Figs. 4, 5)" L482.

**Reviewer comment:** L518: "pattern"

**Author's response:** Spelling corrected "pattern" L523.

**Reviewer comment:** L529: "(e.g., Figs. 4j, 5i)"

**Author's response:** Edited and corrections made "(e.g., Figs. 4j, 5i)" L534.

**Reviewer comment:** L535: "pressure patterns by modulating their positioning which further highlights the dominant role of SAM as mass change driver in East Antarctica", Do you mean "mass change" by "climate" here? Please clarify.

**Author's response:** We meant SAM as a climate driver of mass change. The text has been revised for clarity and now reads: "The SAM phase largely governs these pressure patterns by modulating their positioning, which further highlights the dominant role of SAM as a climate driver of mass change in East Antarctica." This revision appears on L540–542.

**Reviewer comment:** L537: "showed a mass change pattern that is consistent"

**Author's response:** Revised as suggested "showed a mass change pattern that is consistent" L542-543.

**Reviewer comment:** L542: "mass gain in Dronning Maud Land"

**Author's response:** Deleted "the" in the sentence "mass gain in Dronning Maud Land" L548.

**Reviewer comment:** L548: "East Antarctica"

**Author's response:** We have added "a" to the end of Antarctic "East Antarctica" L554.

**Reviewer comment:** L550: Add a period "." At the end of this sentence.

**Author's response:** Period has been added at the end of the sentence "However, in Dronning Maud Land, atmospheric rivers explain about 77 % of interannual variability (Baiman et al., 2023)." L554-556.

**Reviewer comment:** L560: "in our analysis combining La Niña with positive SAM (Fig. 1c)."

**Author's response:** Revised as suggested "in our analysis combining La Niña with positive SAM (Fig. 1c)." L566-567.

**Reviewer comment:** L563: "event likely influenced the mass anomaly … La Niña period."

**Author's response:** The text has been revised "event likely influenced the mass anomaly patterns of the 2020–2022 La Niña period" L569-570.

**Reviewer comment:** L568-569: "events in October 2021 and March 2022 …"

**Author's response:** Date arranged in ascending order "events in October 2021 and March 2022" L575.

**Reviewer comment:** L573: "long-term time series"

**Author's response:** "term" added "long-term time series" L579.

**Reviewer comment:** L589: "cumulative SAM index shows a neutral phase"

**Author's response:** Phase has been replaced with "index" now reads "cumulative SAM index shows a neutral phase" L596.

**Reviewer comment:** L590: "between 2000 and 2020"

**Author's response:** We have replaced "to" with "and" now reads "between 2000 and 2020" L597.

**Reviewer comment:** L598-599: "Regarding the impact of SAM on basal melting, …"

**Author's response:** We have replaced "In terms" with "Regarding" and inserted "of SAM" before on basal melting "Regarding the impact of SAM on basal melting" L605-606.

**Reviewer comment:** L601: "the timescale of the upstream ice response …"

**Author's response:** Revised as suggested "the timescale of the upstream ice response of the upstream to positive" L608-609.

**Reviewer comment:** L627: "(which we removed)"

**Author's response:** "d" has been added to remove ""(which we removed)" L636.

**Reviewer comment:** L651: "which influences its impact on the"

**Author's response:** Revised as suggested "which influences its impact on the Antarctic" L659.

**Reviewer comment:** L653: "…2013), and ENSO has its strongest … West Antarctica. In East Antarctica, atmospheric … moisture influx affecting ice mass variability."

**Author's response:** Revised as accordingly "(Hosking et al., 2013), and ENSO has its strongest impact in West Antarctica. In East Antarctica, atmospheric pressure patterns over the Southern Ocean play a crucial role in regulating moisture influx affecting ice mass variability." L661-663

**Reviewer comment:** L663: "resemble more"

**Author's response:** "more" added "resemble more an" L671.

**Reviewer comment:** L666: "net ENSO effect on AIS mass change as explored here."

**Author's response:** Revised as suggested "net ENSO effect on AIS mass change as explored here" L674.

**Reviewer comment:** L671: "… (2003) are available"

**Author's response:** Inserted "are" on L679: "2003) are available".

**Reviewer comment:** L688: "… (2024) for providing RACMO2.4p1 SMB."

**Author's response:** Dataset deleted and RACMO2.4p1 added before SMB "(2024) for providing RACMO2.4p1 SMB." L696

---

## Author Response (AR4)

We extend our sincere gratitude to the editor for accepting our article for publication in *TC*. The two technical corrections you identified have been fully addressed, and the corresponding revisions have been incorporated into the updated manuscript.

**Technical corrections:**

**Reviewer comment:** L605-606: "However, the timescale of the upstream ice response to positive SAM forcing is unclear …"

**Author's response:** In the revised manuscript, we have deleted "of the upstream" in the sentence and now reads "However, the timescale of the upstream ice response to positive SAM forcing is unclear…." L605-606.

**Reviewer comment:** L666-667: "Although climate model projections remain uncertain regarding whether future ENSO events will resemble more an El Niño- or La Niña-like state…"

**Author's response:** We have revised the text as suggested and deleted the "more" before "resemble" which now reads: "L666-667: Although climate model projections remain uncertain regarding whether future ENSO events will resemble more an El Niño- or La Niña-like state.."